

# Attribute identification based IoT fog data security control and forwarding

Jingxu Xiao[1], Chaowen Chang[1], Ping Wu[1] and Yingying Ma[1,2]

[1] Information Engineering University of the Army Strategic Support Force, Zhengzhou, China
[2] Zhengzhou University of Technology, Zhengzhou, China

## ABSTRACT

As Internet of Things (IoT) applications continue to proliferate, traditional cloud computing is increasingly unable to meet the low-latency demands of these applications. The IoT fog architecture solves this limitation by introducing fog servers in the fog layer that are closer to the IoT devices. However, this architecture lacks authentication mechanisms for information sources, security verification for information transmission, and reasonable allocation of fog nodes. To ensure the secure transmission of end-to-end information in the IoT fog architecture, an attribute identification based security control and forwarding method for IoT fog data (AISCF) is proposed. AISCF applies attribute signatures to the IoT fog architecture and uses software defined network (SDN) to control and forward fog layer data flows. Firstly, IoT devices add attribute identifiers to the data they send based on attribute features. The ingress switch then performs fine-grained access control on the data based on these attribute identifiers. Secondly, SDN uses attribute features as flow table matching items to achieve fine-grained control and forwarding of fog layer data flows based on attribute identifiers. Lastly, the egress switch dynamically samples data flows and verifies the attribute signatures of the sampled data packets at the controller end. Experimental validation has demonstrated that AISCF can effectively detect attacks such as data tampering and forged matching items. Moreover, AISCF imposes minimal overhead on network throughput, CPU utilization and packet forwarding latency, and has practicality in IoT fog architecture.

## INTRODUCTION

The Internet of Things (IoT) (*Guo et al., 2021*) is a technology that connects various physical devices through the network and enables them to collect, transmit and process data. It has extensive applications in areas such as smart homes, transportation, and healthcare, thanks to its principle of universal connectivity. Cloud computing, as a network computing model, has been widely used in the IoT field (*Chen et al., 2021*). However, in practical applications, the cloud data center is far away from the IoT devices, which may cause latency and security risks in data transmission (*Nurmi, 2022*). Especially in large-scale IoT application scenarios, massive data transmission between devices may cause long latency due to network congestion, transmission distance and other factors, affecting

Corresponding author
Jingxu Xiao,
xiaojingxu2301@163.com

real-time performance and user experience. Fog computing (*Costa et al., 2022*), which emerged as a solution to these issues deploys computing, storage and network services closer to the data sources, effectively reducing data transmission latency and improving real-time performance (*Chang et al., 2020*). At the same time, fog computing can also ease the pressure on cloud servers, save network bandwidth and computing resources.

IoT-Fog architecture divides the IoT into three parts: the device layer, the fog layer, and the cloud layer (*Chiang & Zhang, 2017*), as shown in Fig. 1. The device layer handles data collection and local control, while the fog servers within the fog layer undertake data processing and analysis. The cloud layer, on the other hand, manages advanced data services and applications. The IoT-Fog architecture brings efficient computing and storage, but also faces some security threats (*Aleisa et al., 2022*). For example, the security levels of different IoT devices vary greatly, and when the fog layer does not implement defense mechanisms for accessing devices, there is a high risk of low-security-level device access (*Zhang & Zhou, 2018*); illegal malicious IoT devices can inundate fog nodes with false requests, preventing the nodes from receiving and processing regular data (*Kolias, Kambourakis & Stavrou, 2017*); IoT devices can launch deception attacks on fog nodes by forging IP and Mac addresses (*Javed et al., 2021*). When data is transmitted in the fog layer, malicious nodes in the link will steal, tamper with or delete data, affecting the safe use of IoT users (*Kang et al., 2019*). Therefore, ensuring end-to-end data security transmission in the fog layer is the key to IoT data security.

The software-defined network (SDN) is an innovative network architecture introduced by the CleanSlate team at Stanford University (*Kreutz et al., 2014*). Unlike traditional networks, SDN separates the control plane from the data plane, allowing the data plane to focus solely on data forwarding while the control plane manages the flexible control and forwarding of data flow within the network. Programming Protocol-Independent Packet Processors (P4) (*Bosshart et al., 2014*), a state-of-the-art programming language and framework, enables the programmability of forwarding devices, such as switches, in the data plane. By embracing P4, networks can effectively implement the network data control function, as originally envisioned in the concept of SDN. SDN has the characteristics of centralized management, scalability and flexible programmability. By deploying SDN in IoT, it is convenient to efficiently manage massive data flows. At the same time, SDN can realize the detection and security verification of data transmission between IoT devices and fog nodes (*Rafiq et al., 2022*).

*Maji, Prabhakaran & Rosulek (2011)* formally proposed the Attribute-Based Signature (ABS) in 2011. By introducing a signature policy, only signers who meet the attribute set requirements in the policy are allowed to sign, ensuring fine-grained access control of the signer while authenticating the source of information. By applying attribute signatures to the source authentication of IoT devices and generating keys based on attribute features, it can effectively solve problems such as establishing keys for all users and occupying a large amount of key storage space. The attribute features used in attribute signatures can effectively describe data flows. SDN can effectively improve the fine-grained data forwarding in IoT fog architecture by using attribute features as matching items. Meanwhile, attribute

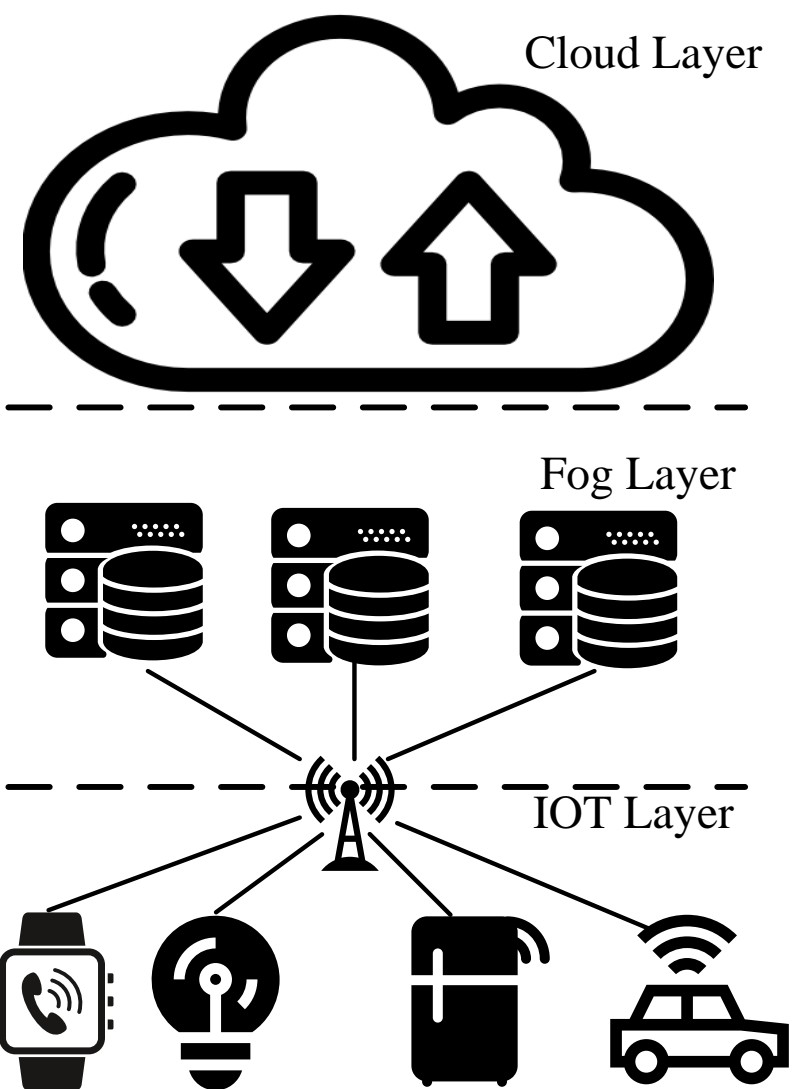

**Figure 1** **IoT fog architecture diagram.**

signatures can effectively guarantee the non-forgery and integrity of data, and realize the security verification function of end-to-end data in IoT fog layer.

To solve the security problems in the current IoT fog architecture and realize the security transmission of end-to-end data in the fog layer, we propose an attribute identification based security control and forwarding method for IoT fog data. It uses attribute signatures to perform access control on IoT devices, adds attribute identity headers to the data flowing into the fog layer, and combines SDN to perform fine-grained forwarding, dynamic sampling and security verification on the fog layer data. The main contributions of this paper are as follows:

(1) We propose a method of data security control and forwarding based on attribute identification, which is called AISCF. AISCF establishes an access control mechanism

based on attribute features for data flows, ensuring the security and traceability of incoming data flows in IoT fog layer. It also uses attribute identification as the basis for SDN control forwarding, enabling fine-grained control forwarding based on attribute characteristics for data flow. Furthermore, it detects malicious packets such as data tampering and matching item forgery based on attribute signature, ensuring the security of end-to-end data packet forwarding process.

(2) We propose a method of dynamic sampling for data flows, which adjusts the sampling factor according to the verification results of data packets, effectively reducing the verification overhead of AISCF.

(3) We test AISCF by simulating the IoT fog architecture environment using P4 software switches. Through experiments, we verify that AISCF is effective. At the same time, AISCF has less overhead in terms of forwarding delay, throughput and controller CPU usage, and is a lightweight scheme suitable for IoT fog architecture.

The rest of this article is organized as follows. The 'Related work' section introduces the related research on IoT fog data forwarding security. The 'Problem description' section summarizes the threat model of end-to-end packet forwarding and proposes the goal of this article. The 'Attribute identification based security control and forwarding' introduces the architecture and the specific implementation of MSLPFV. The 'Analysis and discussion' section analyzes and discusses the security proof, overhead, and implemented functions of AISCF. The 'Experiment and evaluation' section constructs experiments to test and evaluate AISCF. The 'Conclusion' section concludes the article.

## RELATED WORK

To ensure secure data transmission in the IoT fog layer, current security defense measures mainly consist of firewall, anomaly detection, access control and data verification techniques. *Kamoun-Abid et al. (2019)* suggested a firewall architecture based on the integration of cloud and fog layers. As fog nodes are closer to IoT devices, they can filter traffic efficiently according to rules, effectively reducing the hop count of data packets in the IoT. *Sadiq, Thompson & Ayeni (2020)* leveraged Software Defined Networking (SDN) to defend against DDoS attacks in the IoT. Although firewalls incur low overhead, they are unable to defend against internal and unknown attacks effectively. *Dhawan et al. (2015)* introduced Sphinx, which collects traffic information from switches, constructs flow graphs for specific traffic information in the network, and monitors network traffic effectively based on flow graph features. *Nguyen et al. (2019)* employed distributed controllers to obtain IoT traffic information, alleviating the single point of failure problem, and combined deep learning to build three different levels of intrusion detection systems to detect abnormal traffic. However, anomaly detection schemes require real-time monitoring and statistics of IoT traffic, which increases network load when monitoring all data streams and causes false negatives when monitoring specific types of data streams. Moreover, anomaly detection introduces some detection delay, making it hard to block malicious traffic attacks in real time.

By establishing access control and authentication mechanisms for the fog layer traffic, the inflow and circulation of illegal data can be effectively controlled. The Virtual source

Address Validation Edge (VAVE) framework (*Yao, Bi & Xiao, 2011*) added a source address verification module in the controller, which can effectively filter data streams with spoofed addresses. *Li et al. (2017)* modified OpenFlow protocol and Bloom filter to ensure the security of the channel between SDN controller and switch, and to resist man-in-the-middle attacks launched by malicious nodes in the IoT, but the scheme did not describe the method of confirming malicious nodes. *Al Hayajneh, Bhuiyan & McAndrew (2020)* employed TLS/SSL proxy to solve the problem of IoT device reconfiguration. The scheme isolated the traffic from abnormal physical devices by analyzing the source MAC address of the traffic in the control plane. However, the scheme used deep packet inspection technique, which incurred high time overhead. *Muthanna et al. (2019)* introduced blockchain into the IoT, and used SDN controller to authenticate IoT devices, but did not implement blockchain in the scheme. *Gao et al. (2019)* collected flow information from requesters to responders, and built an identity verification model based on blockchain, achieving secure management of IoT fog architecture. *Xie et al. (2019)* applied blockchain technology to vehicular networks. Each vehicle in the system contained a road information, which was rated by its surrounding vehicles for its authenticity, thus ensuring the correctness of the information. Although blockchain can authenticate IoT devices, it generates a lot of information exchange between controllers or fog nodes, and schemes (*Muthanna et al., 2019*; *Gao et al., 2019*; *Xie et al., 2019*) are not suitable for IoT devices that need fast feedback. *ELMansy, Metwally & Badran (2022)* utilized SDN controller's traffic monitoring function, and used MPTCP mode in multipath routing to manage fog nodes securely, allowing redundant communication between nodes, but the scheme had fixed fog node topology structure and required modifying TCP/IP protocol. To enable users to flexibly use data streams on demand, *Halpern & Pignataro (2015)* proposed service function chain (SFC), which built the SDN flow table and controlled and forwarded data streams according to different user needs, but SFC did not analyze the security of the scheme. *Zhu et al.*'s (*2020*) Attribute-Guard used attribute features to classify rules issued by SDN controller into fine-grained categories, improving the drawback of fixed OpenFlow match item categories, and combined digital signature to verify the source of different types of data streams, but the scheme executed signature verification process on switches, introducing large forwarding delay.

Encrypting and verifying data with cryptographic algorithms can effectively enhance the security of end-to-end communication in the IoT fog architecture. *Mohan, Kodati & Krishna (2022)* encrypted flow rules through SDN controller, and used SDN switch as fog layer server to decrypt flow rules, ensuring the integrity of flow rules, but the scheme could not identify abnormal data streams. LPV (*Wang, Li & Zhang, 2019*) calculated the message authentication code of data packets, and compared the message authentication codes of data packets when they passed through the ingress switch and egress switch, ensuring the integrity of data packet forwarding process. However, LPV calculated the message authentication code of data packets through the ingress switch, and could not effectively distinguish data packets sent by illegal visitors. SDNsec (*Sasaki et al., 2016*) added a code containing transmission path information to data packets. When data packets passed through switches in the network, switches verified this field and discarded data packets

that violated path rules, ensuring the consistency of data packet forwarding path in SDN. However, SDNsec required verifying message authentication code of data packets on each switch, which had high configuration requirements for switches in the network. P4Label (*Zuo et al., 2020*), a packet forwarding control mechanism for SDN based on P4, detected attacks such as malicious tampering and data forgery, but P4Label added extra header length to data packets, which was relatively long, and generated large key storage overhead. *Qin, Tang & Chang (2018)* introduced the concept of cryptographic identity into SDN's control forwarding, generating cryptographic identity from user's identity, file attributes or business content and other features, and verifying signature when data packets entered and exited network, ensuring security of data packet transmission, but this method caused problems such as frequent key replacement and large number of keys in implementation process. *Xiao et al. (2022)* proposed a secure data flow forwarding method based on service ordering management, which establishes a corresponding relationship between users, hosts, and services by formulating business rule tables. This method only allows authenticated users and hosts to use the service and performs business-based ordered management of data flows. However, this scheme uses digital signatures to perform security verification on matching items, which can only ensure that the matching items are not tampered with. Attackers can still illegally access the network by stealing and forging legitimate matching items.

Table 1 summarizes the relevant research schemes. From the table, it can be seen that research on establishing secure access control mechanisms for data and secure verification of data are separate. Furthermore, some schemes have issues such as high communication costs, long anomaly detection times, and high verification overheads. Therefore, there is currently a lack of a low-cost method that can balance access control and data security verification for the Internet of Things fog architecture.

## PROBLEM DESCRIPTION

This section presents the security threats encountered during the end-to-end data forwarding process within the IoT fog architecture. It also proposes objectives to address these issues based on the challenges currently faced.

### Attack model

Assuming that an SDN architecture has been established in the IoT, Fig. 2 illustrates the security threats encountered during the data forwarding process in the IoT. The data in the figure originates from IoT devices, and the receiver is a fog node or a fog server. The solid part of the figure depicts the data forwarding process, and the dashed line indicates the control process of the controller to the IoT switch. Lines of different colors denote different types of data streams. The following describes the data streams shown in the figure:

Security dataflow: A legal IoT device sends a data packet to switch A (1). After receiving the packet, switch A checks whether it has the flow table information required for controlling and forwarding the packet. If switch A has the information, it forwards the packet according to the flow table information (4). If switch A does not have the flow table information that matches the packet, it sends a request for flow table information to the controller (2).

**Table 1 Summary of related schemes.**

| Scheme | Security Function | Main Technology | Issues |
|---|---|---|---|
| Scheme (*Kamoun-Abid et al., 2019*; *Sadiq, Thompson & Ayeni, 2020*) | Filtering malicious traffic | Firewall | Unable to defend against internal and unknown attacks |
| Scheme (*Dhawan et al., 2015*; *Nguyen et al., 2019*) | Detecting malicious traffic | Anomaly detection | Monitoring and statistics of traffic cause high overheads, and detection has lag |
| Scheme (*Yao, Bi & Xiao, 2011*; *Li et al., 2017*) | Security of data sources | Access control | Unable to prevent tampered malicious traffic |
| Scheme (*Al Hayajneh, Bhuiyan & McAndrew, 2020*) | Isolating abnormal traffic | Deep packet inspection | Payload detection requires high time overhead |
| Scheme (*Muthanna et al., 2019*; *Gao et al., 2019*) | Device authentication | Blockchain | Multiple nodes record and authenticate information, resulting in high communication overhead |
| Scheme (*Xie et al., 2019*) | Authentication of information authenticity | Blockchain | Large amount of information exchange between nodes, unable to timely block abnormal information |
| Scheme (*ELMansy, Metwally & Badran, 2022*; *Halpern & Pignataro, 2015*) | Flexible control of services | Software-defined network | Security analysis of data is not performed |
| Scheme (*Zhu et al., 2020*) | Source authentication | Digital signature | Verification process is performed on the switch, resulting in high time overhead |
| Scheme (*Mohan, Kodati & Krishna, 2022*) | Flow rule integrity | Encryption | Unable to detect abnormal data |
| Scheme (*Wang, Li & Zhang, 2019*) | Data packet forwarding verification | Sampling verification | Ingress and egress exchange devices need to sample data packets separately, increasing communication overhead |
| Scheme (*Sasaki et al., 2016*) | Consistency verification of forwarding path | Message verification code | All switches need to participate in the verification process, and the header length increases linearly with the path |
| Scheme (*Zuo et al., 2020*; *Qin, Tang & Chang, 2018*) | Data packet forwarding verification | Digital signature | Different types of data need to generate keys, resulting in large key storage overhead |
| Scheme (*Xiao et al., 2022*) | Secure and ordered data flow | Digital signature | Unable to detect illegal data flows with forged matching items |

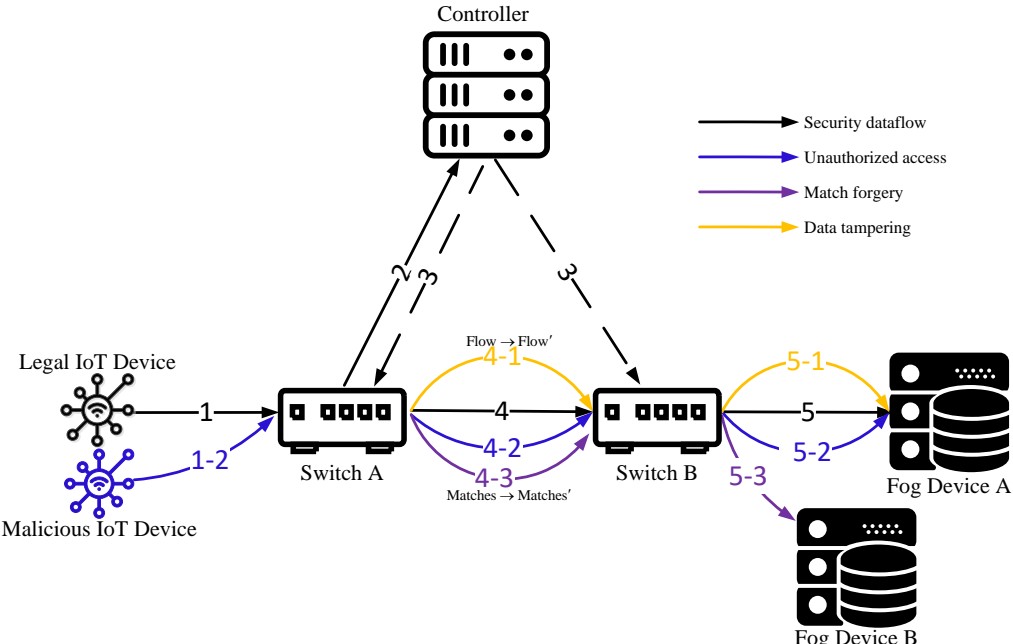

**Figure 2** **Security threats faced by IoT during data flow forwarding.**

Upon receiving the request, the controller sends the corresponding flow table information to switch A and switch B (3), and switch A forwards the packet according to the flow table information (4). After receiving the packet from switch A, switch B performs the same process as switch A and forwards the packet to Fog Device A.

Unauthorized access: Unauthorized IoT devices or users transmit data within the IoT. Unauthorized devices and users refer to those that have not obtained IoT authorization. When an unauthorized device (user) sends a data packet to switch A (1-2), switch A cannot differentiate between authorized and unauthorized devices, and will forward the data packet from the malicious device to Fog Device A according to the same process as for secure data packets (4-2,5-2), thus exposing Fog Device A to attack.

Matches forgery: Malicious nodes in the network forge match items such as IP address, device information, *etc.*, and deceive switching devices to forward data streams abnormally. For example, Switch A is a malicious node (the attacker pollutes the switching device by exploiting vulnerabilities, implanting malicious software, ARP spoofing, *etc.*), and it forges the match item of the data stream and sends it to switch B (4-3). Switch B forwards the data stream to the wrong Fog Device B (5-3) according to the tampered match item.

Data tampering: The malicious nodes in the network tamper with or inject malicious content into the payload of the transmitted packets. For example, suppose that switching device A in the figure is a malicious switching device, which tampers with the received security data packets and forwards them to switch B (4-1). Because the matching items of the data packets tampered with have not changed, switch B forwards the data packets to

Fog Device A (5-1) according to the processing flow of the security data packets. At this time, Fog Device A may be subject to network attacks.

## Objectives

To address the security threats encountered during the end-to-end data forwarding process in the IoT fog architecture, we aim to propose a scheme for secure control and forwarding of data streams. The objectives we hope to achieve with this scheme are as follows:

(1)  To authenticate the origin of data initiators, it is essential to permit only devices that adhere to the access control rules to transmit data within the network. Simultaneously, by integrating the characteristics of data initiators with SDN matching criteria, we can achieve precise access control for data initiators and fine-grained control and forwarding of data flows.

(2)  To detect and block abnormal packets that have data tampering and matching item forgery, and provide the function of tracing malicious data that enter the network, ensuring the security of data transmission.

(3)  To choose an appropriate sampling method for sampling and verifying data flows, it is crucial to avoid creating excessive verification overhead. Such overhead could significantly affect both data transmission efficiency and network performance.

## ATTRIBUTE IDENTIFICATION BASED SECURITY CONTROL AND FORWARDING

To ensure secure data transmission within the IoT fog architecture, we have developed the AISCF model architecture. This section provides an overview of AISCF, covering four key aspects: the overall architecture, the attribute identity structure, the attribute signature scheme, and the AISCF implementation.

### Overall architecture

In AISCF, first, the data initiator needs to obtain the key from the key generation center (KGC) according to its own attribute characteristics, and the attribute identification adding module adds the attribute identification header to the data packet and sends it to the network. The ingress switch judges whether the data packet has attribute identification, and matches and forwards the qualified data packet based on the attribute identification. The intermediate forwarding device only performs matching forwarding based on attribute identification for data packets. The egress switch samples the data packets while forwarding them, and sends the sampled data packets to the controller. The controller performs security verification on the sampled packets based on the attribute signature parameters obtained from the KGC. Figure 3 depicts the AISCF architecture.

The AISCF architecture consists of several components, each serving specific functions:

KGC: The KGC is responsible for generating and distributing keys used in the packet attribute signature phase. Users provide their own attribute set and attribute policy set for the signature. The KGC verifies the attribute policy set, generates the key, and provides users with the attribute private key required for signature. Additionally, the KGC maintains

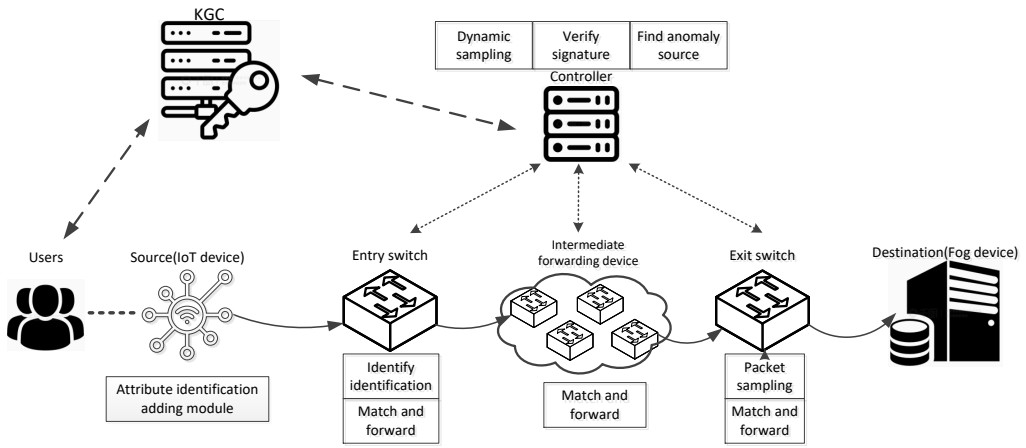

**Figure 3    AISCF architecture.**

communication with the controller, supplying the attribute policy set and system public parameters needed for verifying attribute signatures.

Attribute identification adding module: The attribute identification header is added to the packet to realize control forwarding and security verification of data packets entering the IoT based on attribute identity.

Forwarding device: The forwarding device selects the programmable P4 switch that can effectively identify the attribute identification field and processes its contents. We classify the switches into ingress switches, intermediate forwarding devices, and egress switches based on the packet forwarding order. These switches realize their respective functions according to the multilevel flow table. The ingress switch identifies the attribute identification field of the incoming packets and only forwards the packets with attribute identification according to the flow table information. The intermediate forwarding device only forwards the packets that match based on attribute identification. The egress switch dynamically samples the incoming packets and uploads the sampled packets to the controller, while forwarding the unsampled packets based on attribute identification.

Controller: The controller adopts a dynamic sampling mechanism based on data packet security verification, which dynamically updates the sampling factor according to the verification results of the sampled data packets, and controls the egress switching devices to perform dynamic sampling on the data flows. The controller verifies the signatures on the attribute identifiers of the sampled data packets. In the event of an abnormal data packet failing verification, the controller promptly updates the flow rules based on the attribute identifiers of the abnormal data packets, ensuring the security of the data flow forwarding paths. Through communication with the KGC, the controller can track and confirm the identity of the packet initiator, effectively tracing the source of abnormal packets in the network.

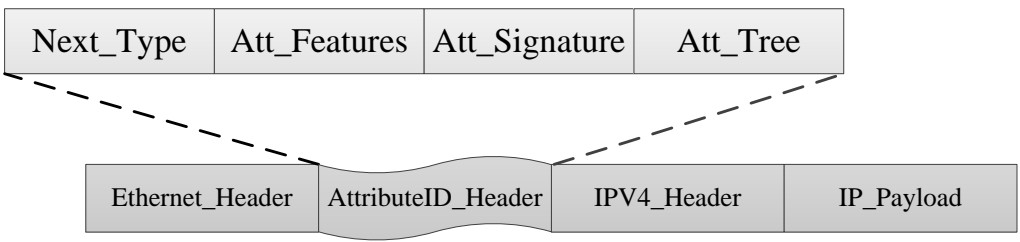

**Figure 4** **Attribute identification structure.**

## Attribute identification structure

The packet header defines the attribute identification field, which is added between the Ethernet header and the IP header in the traditional packet. The attribute identification structure is shown in Fig. 4. The attribute identification field comprises four parts: Next_Type, Att_Features, Att_Signature, and Att_Tree. AISCF can realize secure and fine-grained control and forwarding of dataflow based on attribute identification.

Each part of the attribute identification field is discussed here:

Next_Type (16 bits): It acts as a logical connection between multiple headers. After parsing the attribute identification field, the switch determines the header field that needs to be parsed next through Next_ Type to realize the orderly parsing of various headers.

Att_Features (64 bits): Used to represent the set of packet attribute features. The attribute class features, such as user, device, and service categories, are used to replace deterministic features to describe the dataflow. And Att_Features is generated by calculating the hash function of attribute characteristics. AISCF implements fine-grained control and forwarding of dataflow according to Att_Features.

Att_Signature (256 bits): It is the packet attribute signature field. By using the attribute signature scheme, the Att_Features section and the IP_Payload section of the packet is signed and stored in Att_Signature. AISCF verifies the security of the data packages according to Att_Signature.

Att_Tree (128 bits): It stores the attribute signature policy in a tree structure. Att_Tree contains attribute features and the logical relationship between attribute features. Only the attribute set that conforms to the signature policy can sign the packet. Att_Tree is also used as an input parameter in the packet attribute signature verification phase.

## Attribute based signature scheme

ABS scheme poses problems such as high time consumption and linear correlation between signature verification time and the number of attributes (*Su et al., 2020*). Packets experience large transmission delays when using ABS verification. To address these problems, AISCF extends and employs the ABS scheme proposed by *Tang, Ling & Shan (2022)* in the packet forwarding process through SDN. The scheme incurs low time overhead in the signature verification process and is independent of the number of attributes, enabling

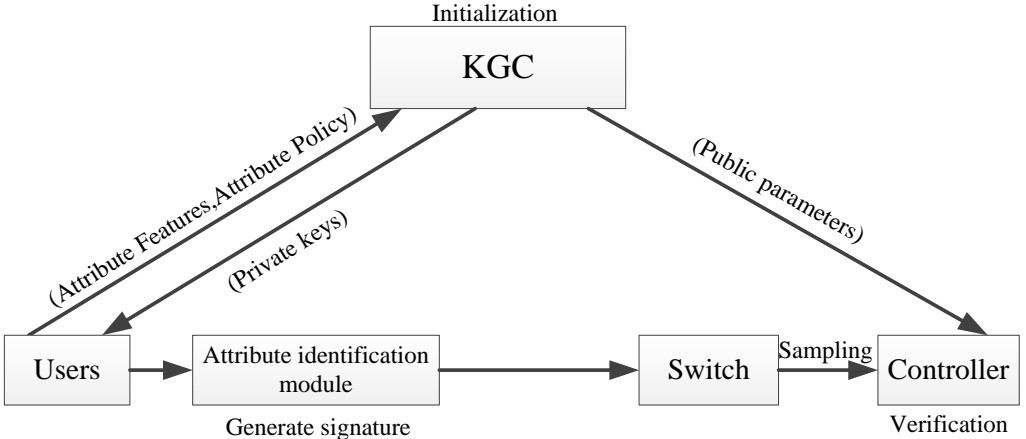

**Figure 5** Application of the proposed ABS scheme in AISCF.

lightweight and fine-grained secure forwarding verification of data packets. Figure 5 shows the implementation process of the attribute signature scheme in AISCF.

AISCF adopts the concept of attribute tree proposed by *Bethencourt, Sahai & Waters (2007)* and uses the tree structure to restrict the access policy of the signer's attribute features. The signer qualifies to sign only when they match the access structure of the attribute policy tree. Threshold and attribute nodes compose the nodes in the attribute policy tree. Each attribute node corresponds to an attribute of the signer, whereas the threshold node represents the structure of $n$ inputs corresponding to one output. For example, $(m, n)$ means that effective output is possible only when at least $m$ inputs out of a total of $n$ inputs satisfy the conditions. In particular, when $m = $n, $(m, n)$ represents the "and" relationship; when $m = $1, $(m, n)$ represents the "or" relationship.

Figure 6 illustrates an example of the attribute policy tree structure. As shown in the figure, the signer with administrator characteristics or the authority to use the authentication host conforms to the signature rules of the attribute policy tree. If the signer has the attribute features of user and Service C, they can effectively sign. However, if the signer has the attribute features of user and Service D, they cannot execute the signature because it does not conform to the access structure of the attribute policy tree.

Figure 6 illustrates an example of an IoT access control structure based on an attribute policy tree. In this structure, users with administrative privileges can access the IoT. Regular users can send data to the IoT *via* authenticated devices, with service types A, B, or C. If a user does not have administrative status, uses a non-authenticated IoT device, or sends other types of services, they will not be able to access the IoT due to non-compliance with the attribute policy tree structure. This structure provides secure and controlled access to the IoT system.

To clearly describe the attribute signature scheme, the following definitions are given.

**Definition 1**. Define the basic parameters in the ABS scheme. $p$ and $N$ are two large prime numbers, $E$ is defined as an elliptic curve over a finite field $F_p$, $G_1$ and $G_2$ are $N$-th

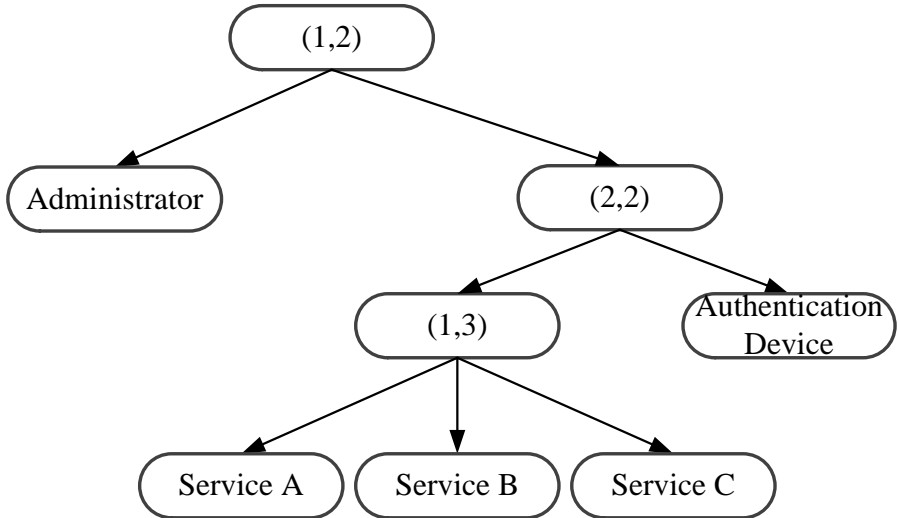

**Figure 6** Attribute policy tree structure.

order additive cyclic groups on the elliptic curve, $G_T$ is an $N$-th order multiplicative cyclic group, and $P_1$ and $P_2$ are, respectively the generators of $G_1$ and $G_2$. $[x]P_i$ is defined as $x$ times of generator $P_i$, where $x \in Z_N^*$. $\varphi$ is defined as a mapping of $G_2 \to G_1$ such that $\varphi(P_2) = P_1$. $H_1$ and $H_2$ are hash functions, and *hid* is the identification used for generating the key for the user. $e$ is defined as a bilinear mapping of $G_1 \times G_2 \to G_T$ and must meet the following conditions:

(1) Bilinearity: $e([a]P_1, [b]P_2) = e([b]P_1, [a]P_2) = e(P_1, P_2)^{ab}$, where $a, b \in Z_N^*$.

(2) Non-degeneracy: $e(P_1, P_2) \neq 1$, where 1 is an identity element of $G_T$.

(3) Computability: For $P_1 \in G_1, P_2 \in G_2$, the value of $e(P_1, P_2)$ can be solved within polynomial time.

**Definition 2**. ID represents the identity information of the signer, and $Att_{\text{ID}}$ represents the attribute set of the signer.

**Definition 3**. $\Upsilon$ represents the access control policy based on attributes, and T represents the attribute policy tree generated based on $\Upsilon$.

**Definition 4**. PP represents the parameters disclosed in the attribute signature process, (msk, mpk) represents the master key pair of the system, in which msk and mpk represent the master public key and the master private key of the system respectively.

**Definition 5**. $m$ represents the signed message, and the hash function H is defined. In AISCF, $m$ is the hash value generated by the Att_Features field in the attribute identification and the data package payload, $m = \text{H}(\text{Att\_Features} \| \text{IP\_Payload})$.

**Definition 6**. sk represents the private key used for signature, and sign represents the attribute signature.

**Definition 7**. The Lagrangian coefficient is defined as $\Delta_i(x) = \prod_{i \in S, j \neq i} \frac{x - x_j}{x_i - x_j}$, where $i \in Z_N^*, S \in Z_N^*$. For the $t - 1$-th order polynomial $p(\cdot)$, any value can be calculated through $t$ given points $(x_i, y_i = p(x_i))$, that is, $p(x) = \sum_{i=1}^{t} p(i) \Delta_i(x) \bmod N$.

The proposed ABS scheme is discussed in terms of five aspects: system initialization, key generation, signature, signature verification, and tracing.

**System initialization** $(1^l) \rightarrow (\text{PP}, \text{mpk}, \text{msk})$: The KGC generates the system master private key msk according to the security parameter $l$, and calculates the system master public key $\text{mpk} = [\text{msk}]P_2$. KGC saves msk and makes mpk public. $g = e(P_1, \text{mpk})$, KGC obtains the public parameters $\text{PP} = (N, P_1, P_2, e, \varphi, G_1, G_2, G_T, hid, g, H_1, H_2)$ according to the basic parameters in Definition 1.

**Key generation** $(\text{msk}, \Upsilon, Att_{\text{ID}}) \rightarrow (\text{sk})$: The user provides the KGC with attribute policy $\Upsilon$ and user attribute feature set $Att_{\text{ID}}$. The KGC then generates the user private key according to the following steps:

(1) It is determined whether the attribute set provided by the user meets the access control policy. $\Upsilon(Att_{\text{ID}}) = 1$ indicates that the user meets the access control policy, and perform step (2); otherwise, $\perp$ is returned, and the user cannot obtain the key.

(2) A random number $t$ is selected. Let $h_{ID} = H_1(\text{ID} \parallel hid, N)$. $\text{sk}' = \left[\frac{\text{msk}}{h_{\text{ID}} + \text{msk}}\right]P_1$ and $\text{sk}_2 = \left[\frac{h_{\text{ID}} + \text{msk}}{t}\right]P_2$ are calculated.

(3) The attribute policy tree T is constructed from $\Upsilon$, and the $k_x - 1$-th order polynomial $p_x(\cdot)$ for each node $x$ in T is generated using the top-down method, where $k_x$ is the threshold of the node. The constant term of the root node is defined as $t$, that is, $p_{x_r}(0) = t$. For a non-leaf node $x'$, $p_{x'}(0) = p_{par(x')}(ind(x'))$, where $ind(x)$ and $par(x)$ represent the numbers of node $x$ and its parent node, respectively. For a leaf node $x''$, the attribute key $\text{sk}_1 = \{\text{sk}_{1,i} = p_{x''}(0) \cdot \text{sk}'\}$, $i = \text{att}(x'')$ is calculated; $\text{att}(x)$ represents the number of node $x$.

(4) The KGC saves $(\text{ID}, \text{sk}_2)$ and returns $\text{sk} = (\text{sk}_1 : \{\text{sk}_{1,i}\}, \text{sk}_2)$ to the user.

**Signature** $(m, \text{sk}, \Upsilon) \rightarrow (\text{sign})$: The user signs message $m$ by using sk, selects a random number $r$, and calculates $w = g^r$, $h = H_2(m\|w, N)$, $k = \text{sk}_2$, $l = (r - h)\text{mod}N$, and $s_i = [l]\text{sk}_{1,i}$, obtaining the signature as $\text{sign} = (h, s_i, k)$. In AISCF, sign is stored in Att_Signature in the attribute identification.

**Signature verification** $(m, \text{sign}, \text{T}) \rightarrow (1/\perp)$: The controller in AISCF will verifie the Att_Signature field in the attribute identification of packets. The signature verification process is as follows:

(1) The Lagrangian interpolation formula is used to calculate $s_{x_r}$ of the root node from bottom to top, that is, $[l]p_{x_r}(0) \cdot \text{sk}'$. When $k_{par(x)} = 1$ for node $x$, $s_{par(x)} = s_x$; otherwise $s_{par(x)} = \sum_{x_i \in bro(x)} s_{x_i} \Delta_{x_i}(x)\text{mod}N$. Finally, $s_{x_r} = \left[\frac{l \cdot t \cdot \text{msk}}{\text{msk} + h_{ID}}\right]p_1$ is obtained. The parameter $x$ is omitted and let $s_r = s_{x_r} = \left[\frac{l \cdot t \cdot \text{msk}}{\text{msk} + h_{ID}}\right]p_1$.

(2) $w' = e(s_r, k) \cdot g^h$ and $h' = H_2(m\|w', N)$ are calculated. If $h' = h$ the signature verification is successful and 1 is returned. Otherwise, the signature verification fails and $\perp$ is returned. Because $w = g^r$, the condition for the success of signature verification is $w' = g^r$. The correctness of the signature verification process can be determined using Eq. (1):

**Table 2  Headers defined by AISCF in P4.**

**Structure of headers in P4 Switch**

```
header InPort_Header{
 fields{
 Next_Type : 16;
 In_Port : 2;
 }
 }
header AttributeID_Header{
 fields{
 Next_Type : 16;
 Att_Features : 64;
 Att_Signature : 256;
 Att_Tree : 128;
 }
 }
```

$$
\begin{aligned}
w' &= e(s_r, k) \cdot g^h \\
&= e\left(\left[\frac{l \cdot t \cdot \mathrm{msk}}{\mathrm{msk} + h_{ID}}\right] P_1, \left[\frac{h_{ID} + \mathrm{msk}}{t}\right] P_2\right) \cdot g^h \\
&= e([r - h] P_1, [\mathrm{msk}] P_2) \cdot g^h \\
&= g^{r-h} \cdot g^h \\
&= g^r
\end{aligned}
\tag{1}
$$

**Tracing** $(\mathrm{sign}) \to (\mathrm{ID})$: It is known that $k = \mathrm{sk}_2$ in $sign(h, s_i, k)$ and the KGC saves $(\mathrm{ID}, \mathrm{sk}_2)$ when distributing the key to the user, then the KGC can trace the signer ID by checking the table of $\mathrm{sk}_2$.

## Implementation of AISCF

This section introduces the implementation of AISCF by explaining the workflows of both the switch and the controller.

### *Switch workflow*

P4 programmable switch can add, read and delete headers in data packets. In addition to Ethernet header, IP header and other traditional packet headers, AISCF defines two header types in P4 switches in the network, namely InPort_Header and AttributeID_Header. The format of the header is shown in Table 2. The unit of the header field length in the table is bit.

InPort_Header is used to identify the source of data packets. When In_Port=01 in InPort_Header, it indicates that the data packet is from the IoT device; when In_Port=00, it indicates that the data packet is from the switch. AttributeID_Header represents the attribute identification header defined in the 'Attribute identification structure' section. The switch controls and forwards the data packets according to the content of AttributeID_Header.

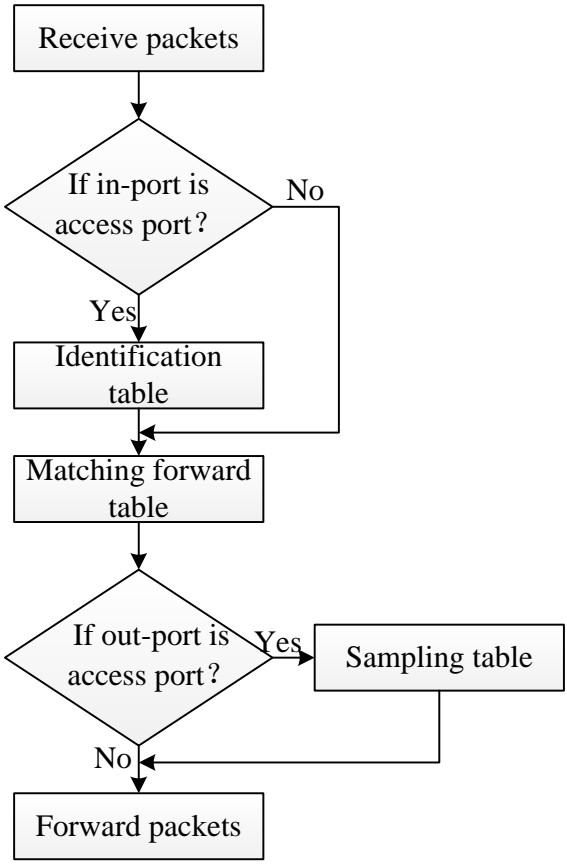

**Figure 7   Multilevel flow matching process.**

The P4 switch will mark the header type for different types of headers. The Next_Type field in the header records the next header type that needs to be parsed. The P4 switch implements the orderly analysis of multiple headers according to Next_Type. The default IP header and Ethernet header type numbers in P4 switch are 0x0800 and 0x1212, and the InPort_Header and AttributeID_Header type numbers are 0x2828 and 0x1818 defined in AISCF. The source host adds the AttributeID_Header, while the ingress switch adds the InPort_Header. The logical position of InPort_Header is in front of the Ethernet header, Next_Type =0x1212 in InPort_Header. AttributeID_Header is in front of the IP header, and Next_Type =0x0800 in AttributeID_Header.

The switch works by executing a "match–action" process on the packet, based on the flow table information it stores and receives from the controller. The switch matches the information parsed from the packet header with the corresponding flow table entry, and then executes the corresponding action on the packet. AISCF architecture establishes port distinguishing table, header validation table, matching forwarding table and sampling table

in P4 switch, and carries out multilevel flow table matching process based on attribute identification for data packets. Figure 7 illustrates the multilevel flow matching process.

As can be seen from Figure, the switch first performs the "match–action" process based on the port matching table for the received packets and executes the corresponding actions for the various input port types. When the input port is an access port, the switch performs the "match–action" process of the identification table for the packets. Next, according to the content of the matching forwarding table, the packet is controlled and forwarded based on attribute identification, and the next hop address of the packet is set. Before the packet flows out of the switch, it is determined whether the exit port of the packet is an access port. If so, the contents of the sampling table are executed. The flow tables shown in Fig. 7 are discussed as follows:

Port matching table: Perform the corresponding action on the packet by matching the switch port type. AISCF divides switch ports into access ports and routing ports based on connection types. The access port indicates that the port is connected to the IoT device, and the routing port indicates that the port is connected to the switch. The port matching table is divided into inbound and outbound port matching tables. In the inbound port matching table, the matching item is In_Port in InPort_Header, when In_Port=01, submit the data packet to the identification table for processing, and when In_Port=00, the switch submits the data packet to the basic forwarding table for processing. In the outbound port matching table, the matching item is standard_metadata.egress_spec, when standard_metadata.egress_spec =01, submit the data packet to the sampling table for processing, and when standard_metadata.egress_spec =00, perform the forwarding action on the packet.

Identification table: Perform the corresponding action on the packet by matching the Next_Type field in the Ethernet_Header. P4 switch extracts the header information from the data packet. When Next_Type =0x0800, it indicates that Ethernet_Header is adjacent to IP_Header, the data packet does not contain attribute identification, and the switch discards the data packet; When Next_Type =0x1818, it indicates that the data contains attribute identification, and the data packet is sent to the matching forwarding table for subsequent processing.

Matching forwarding table: The forwarding policy is formulated based on Att_Features in attribute identification to realize fine-grained packet control and forwarding. The matching and forwarding table matches Att_Features precisely with the type of exact. When Att_Features matches successfully, the switch assigns the parameter standard_metadata.egress_spec based on the matching and forwarding table, where standard_metadata.egress_spec represents the outbound port value of the packet, and forwards the packet, that is, forwards the packet according to the requirements of the matching and forwarding table. When the matching fails, the switch drops the data package and upload the package to the controller through packet-in.

Sampling table: AISCF defines the packet sampling parameter pkt_sample_val in the P4 switch, and the sampling table takes pkt_sample_val as the matching item to sample the packet. When pkt_sample_val =1, the switch executes action clone_ingress_pkt_to_egress

on the data packet and sends the mirror of the data packet to the controller; When pkt_sample_val $=0$, the switch does not process the data packet.

The switch in AISCF processes the data packets according to the Multilevel flow table matching process shown in Fig. 7. Algorithm 1 shows the control and forwarding process of the switch for data packets.

---

**Algorithm 1. Control and forwarding process of the switch for packets**
Input : packet $A$, sampling factor $\theta$
Output : action executed on $A$
1. Function Control and forwarding of packets in switches :
2.   If $A$ has no InPort_Header :
3.     add_header($A$, InPort_Header, In_Port $= 1$)
4.   $in\_port$ $= A$.InPort_Header.In_Port
5.   If $in\_port = 1$ :
6.     $next\_type = A$.InPort_Header.Next_Type
7.     If match($next\_type$, Identification_table) $=$ NULL :
8.       drop($A$)
9.     $A$.InPort_Header.In_Port $= 0$
10.   $att\_features = A$.AttributeID_Header.Att_Features
11.   If match($att\_features$, Matching_forward_table) :
12.     action($att\_features$, Matching_forward_table)
13.     If standard_metadata.egress_spec is Access port :
14.       $sample\_val = $ modify_field_rng_uniform($0, 1 + \theta$)
15.       If $\theta < sample\_val < \theta + 1$ :
16.         pkt_smaple_val $= 1$
17.       Else :
18.         pkt_smaple_val $= 0$
19.       If match(pkt_smaple_val, Sampling_table) :
20.         action(pkt_smaple_val, Sampling_table)
21.       delete_header($A$, InPort_Header, AttributeID_Header)
22.   Else :
23.     drop($A$)
24. END Function

---

In the Algorithm 1, $\theta$ represents the sampling factor, add_header() and delete_header() are defined as functions to add and delete headers to packets, match(*table*, *matching field*) and action(*table*, *matching field*) represent the matching result and execution action between *matching field* in packets and the corresponding flow table *table* respectively, and drop() represents that the switch discards packets.

For data packets entering SDN, the incoming switch adds InPort_Header() to them and performs matching and action execution based on identification table. The switch then sets In_Port in InPort_Header to 0 to inform downstream switches that the routing port sends the data packet (lines 2-9 of Algorithm 1).Then the switch performs the matching and action execution on the packet based on the matching forwarding table. When the switch is an egress switch, the egress switch selects the data packet according to the probability of $1 : \theta + 1$ through the random function modify_field_rng_uniform() in P4 system, and stores the sampling result in pkt_smaple_val (lines 13-18 of Algorithm 1), then performs matching and action execution on the data packet based on the sampling table. Finally,

the egress switch deletes InPort_Header and AttributeID_Header from the packet before forwarding it.

### Controller workflow

AISCF architecture implements the functions of dynamic sampling of data flows, signature verification based on attribute identification for sampled packets, and source tracing of abnormal data packets at the controller end. The realization of controller function in AISCF is shown in Algorithm 2.

---

**Algorithm 2. Processing flow of data package by controller.**
Input : packet $A$, sampled packets number $k$, sampling factor $\theta$
Output : new $k$ and $\theta$, or malicious user identity $ID$
1.Function Security verification, adaptive sanpling and exception tracking of packets by controller :
2.    If $A$ is from switch port :
3.        $m = \mathrm{H}\big(A.\mathrm{AttributeID\_Header.Att\_Features} \| A.\mathrm{IP\_Payload}\big)$
4.        $signature = A.\mathrm{AttributeID\_Header.Att\_Signature}$
5.        $T = A.\mathrm{AttributeID\_Header.Att\_Tree}$
6.        If $\mathrm{verify}(m, signature, T) \neq 1$ :
7.            $\mathrm{drop}(A)$ and $\mathrm{reset\_flow\_path}(A.\mathrm{AttributeID\_Header.Att\_Features})$
8.            $k = 0,\ \theta = \theta_0$
9.        Else :
10.            $k = k + 1$
11.            If $k = \mathrm{k}$ :
12.                $k = 0, \theta = f(\theta)$
13.            return $(k, \theta)$
14.    If $A$ is from controller port :
15.        $signature = A.\mathrm{AttributeID\_Header.Att\_Signature}$
16.        $ID = \mathrm{trace}(signature)$
17.        return $(ID)$
18.END Function

---

In combination with Algorithm 2, the functions implemented by the controller in AISCF architecture are introduced as follows:

**Validate packets based on attribute identification**. When the controller receives the data packet sampled by the egress switch, it generates the signed message $m$ using the same hash function H as the attribute identification adding module. The verify() function verifies the signature of the data packet according to the signature verification process in Section 4.3. When the return value is not 1, it indicates that the packet has anomalies such as matching item forgery or data tampering, which implies that there are malicious nodes in the forwarding path that attack the packet. The controller drops $A$, uses the reset_flow_path() function to reset the forwarding path for the data flow with Att_Features as its attribute feature, and checks for malicious nodes in the old path to ensure the security of the data flow forwarding process (lines 2 to 7 of Algorithm 2).

**Dynamically sampling data flows**. Let $K$ be the number of samples, and the controller adjusts $\theta$ based on the verification result of $A$. If the verification fails, $K$ and $\theta$ are reset to their initial values of 0 and $\theta_0$, and if the verification succeeds, the number of verifications is updated. When $K$ reaches the preset maximum number of samples K, it means that the data flow is securely transmitted. Use the monotonic increasing function $f$ () to update $\theta$,

and decrease the sampling rate of the data flow by increasing $\theta$ (lines 8 to 13 of Algorithm 2). AISCF samples the data flow by updating the sampling factor, which can effectively reduce the verification overhead and improve the transmission performance of the data stream while verifying its security.

**Traceability of abnormal data packets**. When the anomaly detection system or other security defense mechanisms find abnormal packets in the network, the controller in AISCF will effectively locate the sender of the abnormal packets. The controller extracts the Att_Signature field in the attribute identification of the abnormal data packet, and function trace() obtains *ID* of the originator of the abnormal data packet according to the tracking process in the 'Attribute based signature scheme' section, effectively realizing the traceability function of the abnormal data packet. The controller ensures the security of incoming packets in the network by restricting or updating the access control permissions of users with the identified *ID* (lines 14 to 17 of Algorithm 2).

## ANALYSIS AND DISCUSSION

This section will analyze and discuss AISCF from the perspectives of security proof, implemented functions, and scheme overhead.

### Security proof

The security of the cryptographic algorithm is based on difficult mathematical problems. The security of AISCF is based on the complexity assumption of q-SDH (q-Strong Diffie-Hellman Problem) (*Boneh & Boyen, 2004*), For a given $q + 2$ tuple $\left(P_1, P_2, [a]P_2, [a^2]P_2, \ldots, [a^q]P_2\right)$ that is known, it is difficult to solve $\left(x, [\frac{a}{a+x}]P_1\right)$ where $a, x \in Z_N^*, P_1 \in G_1, P_2 \in G_2$. Next, we will prove the unforgeability of attribute signature scheme in AISCF.

We use the method of disproof to demonstrate that if attacker can forge the attribute signature in AISCF with a non negligible advantage $\varepsilon$, then challenger can be constructed to solve q-SDH with a non negligible advantage. Build a q-SDH instance, $\left(P_1, P_2, [a]P_2, [a^2]P_2, \ldots, [a^q]P_2\right)$, in which $a = $ msk, for challenger based on the attribute signature scheme in Section 4.3. Define a binary group $L_2$, and the format of each element in $L_2$ is $(\text{ID}_i, \beta_i)$. Assume that the number of times attacker can obtain $H_1(\text{ID} \parallel hid, N)$ by asking is $t_h$. When attacker asks once for the hash of $\text{ID}_i$, challenger randomly selects $\beta_j$ from $L_2$ and returns it to attacker, $\beta_j \in Z_N^*, j \in [1, t_h]$. Challenger randomly selects $\beta_i$ by $q - 1$ times, $i \in [1, q-1]$, and put the $\text{ID}_i$ corresponding to $\beta_i$ into $L_2$. Challenger generates polynomial $f(a) = \prod_{i=1}^{q-1}(a + \beta_i) = \sum_{i=0}^{q-1} c_i a^i$, where $c_0, c_1, \ldots, c_{q-1} \in Z_N^*$. Let $P_2' = [f(r)]P_2, P_1' = \varphi(P_2'), P_1'$ and $P_2'$ are respectively the generators of $G_1$ and $G_2$, $\text{mpk} = [a]P_2' = \sum_{i=0}^{q-1} c_i a^i$, and make mpk public. The public parameters of challenger, $\text{PP} = \left(N, P_1', P_2', e, \varphi, G_1, G_2, hid, g, H_1, H_2\right)$.

**Key inquiry process**. Define a triplet group $L_3$ of challenger, and the format of each element in $L_3$ is $(\text{ID}_i, \text{sk}^{i'}, \text{sk}^i)$. Assume that attacker can ask $\text{sk}^i$, the key of $\text{ID}_i$, by q times, where $\text{ID}_i$ meets $\Upsilon\left(Att_{\text{ID}_i}\right) = 1$. If $\text{ID}_i$ is not in $L_2$, the query fails. If $\text{ID}_i$ is in $L_2$, challenger constructs a polynomial $f_i(a) = \frac{f(a)}{a+\beta_i} = \sum_{i=0}^{q-2} d_i a^i$, where $d_0, d_1, \ldots, d_{q-2} \in Z_N^*$. $\text{sk}^{i'}$ can be

obtained by Eq. (2):

$$
\begin{aligned}
\mathrm{sk}^{i'} &= \left[\frac{\mathrm{msk}}{h_{\mathrm{ID}_i}+\mathrm{msk}}\right]P_1' = \left[\frac{a \cdot f(a)}{h_{\mathrm{ID}_i}+\mathrm{msk}}\right]P_1 \\
&= [a \cdot f_i(a)]P_1 \\
&= [a \cdot \sum_{i=0}^{q-2} d_i a^i]P_1 \\
&= \sum_{i=0}^{q-2}[d_i a^{i+1}]P_1 \\
&= \sum_{i=0}^{q-2} d_i \varphi([a^{i+1}]P_2)
\end{aligned}
\tag{2}
$$

For leaf node $x''$ in the attribute policy tree, $j = \mathrm{att}(x'')$, then get $\mathrm{sk}_1^i = \mathrm{sk}_{1,j}^i = p_j(0) \cdot \mathrm{sk}^{i'}$, $\mathrm{sk}_2^i = [t \cdot ([\beta_i]P_2' + \mathrm{mpk})^{-1}]^{-1}$. Challenger returns $\mathrm{sk}^i = \left(\mathrm{sk}_1^i : \left\{\mathrm{sk}_{1,j}^i\right\}, \mathrm{sk}_2^i\right)$ which is the key of $\mathrm{ID}_i$ to attacker, and saves $\left(\mathrm{ID}_i, \mathrm{sk}^{i'}, \mathrm{sk}^i\right)$ in $L_3$.

**Signature inquiry process**. Assume that attacker can ask for signature by $t_s$ times, and attacker provide $(\mathrm{ID}_i, m_i)$ at each inquiry. If $\mathrm{ID}_i$ is not in $L_3$, challenger returns randomly generated $\mathrm{sk}^i$ to attacker. If $\mathrm{ID}_i$ is in $L_3$, challenger returns $\mathrm{sign} = (h, s_j, k)$, the signature generated by $\mathrm{sk}^i$ for $m_i$, to attacker.

**Challenge process**. Challenger executes signature verification algorithm on $\left(\mathrm{m}^*, \mathrm{ID}^*, \mathrm{sign}^* = (h^*, s_j^*, k^*), \mathrm{T}\right)$ input by attacker. If $\mathrm{ID}_i$ is not in $L_2$ and $L_3$, the challenge fails. Otherwise, we can get $\mathrm{sk}^{*'} = [\frac{a}{h_{\mathrm{ID}^*}+a}]P_1' = [\frac{a \cdot f(a)}{h_{\mathrm{ID}^*}+a}]P_1$, and because $a \cdot f(a) = \sum_{i=0}^{q-1} c_i a^{i+1}$, through the transformation by long division, we can get $a \cdot f(a) = \gamma(a)(a + h_{\mathrm{ID}^*}) + \gamma_{-1}$, where $\gamma(a) = \sum_{i=0}^{q-2} \gamma^i \cdot a^{i+1}$, $\gamma_{-1} \in Z_N^*$. Then Eq. (3) can be obtained:

$$
\begin{aligned}
\left[\frac{1}{h_{\mathrm{ID}^*}+a}\right]P_1 &= \frac{1}{\gamma_{-1}}(\mathrm{sk}^{*'} - \gamma(a)P_1) \\
&= \frac{1}{\gamma_{-1}}(\mathrm{sk}^{*'} - \sum_{i=0}^{q-2}[\gamma^i \cdot a^{i+1}]P_1) \\
&= \frac{1}{\gamma_{-1}}(\mathrm{sk}^{*'} - \sum_{i=0}^{q-2} \gamma^i \cdot \varphi(a^{i+1}P_2))
\end{aligned}
\tag{3}
$$

Challenger can get $h_{\mathrm{ID}^*}$ By looking up $L_2$, and $(h_{\mathrm{ID}^*}, \frac{1}{\gamma_{-1}}(\mathrm{sk}^{*'} - \sum_{i=0}^{q-2} \gamma^i \cdot \varphi(a^{i+1}P_2)))$ can be used as the solution of challenger to q-SDH.

To sum up, the condition for challenger to challenge successfully is that the three processes of key inquiry, signature inquiry and challenge need to be successful, the probability of success of the three processes are $\left(\frac{q-2}{t_h}\right)^{q-2}$, $\left(\frac{t_s}{t_h}\right)^{t_s}$ and $\frac{1}{t_h}$, and the probability of successful challenge is the product of the three. Then it is concluded that if attacker can forge the attribute signature in AISCF with a non negligible advantage $\varepsilon$, challenger can solve q-SDH with a non negligible advantage $\varepsilon' = \frac{(q-2)^{q-2} \cdot t_s^{t_s}}{(t_h)^{t_s+q-1}} \cdot \varepsilon$. It proves that the attribute signature scheme in AISCF can not be forged, that is, AISCF has security.

## Implemented functions

This section analyzes the safety functions of AISCF based on the safety problems described in 'Attack Model' section.

**Defend against unauthorized access**. AISCF only allows users that meet attribute access control policy $\Upsilon$ to access the network. Invalid users cannot obtain the key from KGC due to their inconsistent attributes, and thus cannot generate attribute identifications. AISCF requires only the ingress switch to check the data packets and discard those that do not contain attribute identifications, effectively preventing the data streams sent by illegal visitors. The security proof in the 'Security proof' section shows that the attribute identification cannot be forged. When the illegal visitor wants to forge the attribute identification, because the illegal visitor does not have the key required for signature, he can only randomly generate the 256bit signature field, and the probability of the generated signature passing the verification is $(2^{256})^{-1}$, which would cause great difficulties for illegal visitors. Therefore, we believe that illegal visitors without keys will not randomly generate attribute identifications.

**Prevent matches forgery and tampering with packet content**. The signature field in the attribute identification is the signature of the hash value of the matching item field and the load field of the packet. The signature ensures the integrity of the information. If someone forges the matching item or tampers with the content of the data package, it will damage the integrity of the information. As a result, it cannot pass the security verification process in the controller based on the attribute signature field.

**Source authentication**. AISCF ensures that packets are sent from trusted sources that conform to attribute access policies by adding attribute identifiers to packets. AISCF has anonymity for the data source, that is, the attribute identification of the data package only contains attribute characteristics, signatures generated based on attribute characteristics, and attribute policy trees. In order to prevent attackers from using the anonymity of attribute signatures to perform illegal operations, AISCF provides the traceability function of data sources, which can trace signer ID through $k$ in signature $\text{sign} = (h, s_i, k)$.

**Fine-grained access control**. AISCF implements fine-grained access control on data flow based on attribute characteristics. The fine-grained access control of AISCF on data flow is determined by the number of attribute characteristics in attribute identification and the structure of attribute policy tree in Att_Tree. Suppose there are $Nt$ attributes of different categories, and $n_t$ represents the number of attribute features in the $t$- th attribute, where $t \in [1, Nt]$. In theory, AISCF can implement up to $\prod_{t=0}^{Nt} n_t$ different access controls on data flows. AISCF can achieve access control of IoT fog architecture under various requirements by flexibly setting the access control structure through attribute policy trees. This effectively solves illegal access behavior caused by low access control granularity.

**Fine-grained forwarding of data flows**. AISCF uses Att_Features field in attribute identifier as a matching term for data stream forwarding, and SDN is used to achieve attribute based forwarding of the data stream, effectively improving the fine-grained control of data forwarding in the IoT fog architecture.

**Table 3  Comparison of overheads among different schemes.**

|  | Extra header overhead | Verification overhead |
|---|---|---|
| SDNsec (*Sasaki et al., 2016*) | $(22+8l)$B | $(l+1)$M |
| P4Label (*Zuo et al., 2020*) | 268B | $3E$ |
| Attribute-Guard (*Zhu et al., 2020*) | 40B | $(|S_\Upsilon|+6)E$ |
| AISCF | 58B | $E+\exp$ |

## Scheme overhead

This section analyzes the overhead of AISCF from two aspects: header overhead and computational complexity.

**Additional header overhead**. AISCF realizes its function by adding AttributeID_Header header to the data packet. The length of AttributeID_Header added to the data packet at the source host is 464 bit (58B).

**Computational complexity**. exp is defined as the exponential operation on $G_T$; $sca_1$ and $sca_2$ are the scalar multiplication operations on $G_1$ and $G_2$, respectively; $E$ represents a bilinear mapping operation; and $|S_\Upsilon|$ is the number of attribute features in attribute policy $\Upsilon$. The time cost of the scheme in the key generation phase, signature phase, and signature verification phase is $|S_\Upsilon+1|sca_1+sca_2$, $|S_\Upsilon|sca_1+\exp$, and $E+\exp$, respectively. The computational complexity of the tracing process is $O(n)$, where $n$ is the number of users stored in AISCF. AISCF exhibits a linear correlation between the time complexity and the number of attributes during key generation and signature. The key generation stage only needs to occur once for the same user, while the attribute signature stage takes place at the source device side, not affecting the forwarding delay of packets in the network.

Table 3 compares AISCF with related schemes in terms of extra header overhead and the verification overhead of data packets during the forwarding process. In the table, $l$ denotes the number of switches in the forwarding path, and M denotes the computation process of a message authentication code. It can be seen from Table 3 that the extra header overhead and the verification overhead of data packets for SDNsec (*Sasaki et al., 2016*) grow linearly with $l$, which will incur large overhead when the forwarding path is long; P4Label (*Zuo et al., 2020*) introduces large extra header overhead and requires three pairing bilinear operations at the egress switch; Attribute-Guard's (*Zhu et al., 2020*) verification overhead of data packets increases linearly with the number of attribute features, when the value of $|S_\Upsilon|$ is large, it will cause huge cost to the controller. Compared to related schemes, AISCF incurs less extra header overhead and has smaller and more stable time overhead for data packet verification.

Additionally, AISCF employs dynamic sampling to sample the data flow, which reduces the verification overhead incurred by sampling and verifying the data flow by dynamically changing the sampling factor. Next, we compare the verification overhead incurred by using dynamic sampling and using a fixed sampling factor. Assume that the data stream contains N packets, and the initial fixed sampling factor is $\theta_0$. In dynamic sampling, when K consecutive packets are verified successfully, use $f$ () to update the sampling factor, which

is represented by $\theta_i$. For ease of computation, let N have the value as shown in Eq. (4), which means that the sampling factor is changed $n$ times in the dynamic sampling process of N packets.

$$N = \sum_{i=0}^{i=n-1} K(1+\theta_i). \tag{4}$$

Using the number of packets that are sampled and verified from the data flow as a measure of the verification overhead, when a fixed $\theta_0$ is used to sample and verify the data flow, the number of packets that are sampled and verified from the data stream is $n_0 = N/(1+\theta_0)$; when dynamic sampling is used, the number of packets that are sampled and verified from the data stream is $n_1 = nK$, then the ratio of $n_1$ to $n_0$ can be calculated as shown in Eq. (5).

$$\begin{aligned}
\frac{n_1}{n_0} &= \frac{nK(1+\theta_0)}{N} \\
&= \frac{nK(1+\theta_0)}{\sum_{i=0}^{i=n-1} K(1+\theta_i)} \\
&= \frac{n+n\theta_0}{n+\sum_{i=0}^{i=n-1}\theta_i}.
\end{aligned} \tag{5}$$

Where $\theta_i = f(\theta_{i-1})$, and $f$ () is a monotonic increasing function, so for any $\theta_0 < \theta_i$, and it can be concluded that for the same data flow, the dynamic sampling method produces less verification overhead than the fixed sampling method.

# EXPERIMENT AND EVALUATION

In this section, we establish an experimental environment and evaluate the validity and performance of AISCF through a series of experiments.

## Experimental environment

The AISCF architecture is implemented by employing P4 programmable software switches. These switches utilize the P4 language and generate a JSON format description file *via* the P4 compiler (P4c). Subsequently, this description file is imported into P4 behavioral-model version 2(BMv2) for execution. Additionally, we enhance security by incorporating an attribute signature verification function in Python within the controller. The controller communicates with the data plane through the P4 Runtime interface.

We conducted the experiment on an Intel i7–11370H 4.266-GHz host with 32-GB memory. We used mininet to simulate the IoT fog architecture environment, and Fig. 8 shows the network topology, including two hosts Host1 and Host2 (simulating IoT devices and IoT fog layer servers respectively), four P4 software switches Switch1 to Switch4, and one controller. We generated the data message required for the experiment using scapy and sent it from the host. We also captured and analyzed the operation flow information using Wireshark.

## Validity analysis

**Experiment 1**: We conducted Experiment 1 to verify whether AISCF can realize access control, security verification, and fine-grained control forwarding of data flow, targeting

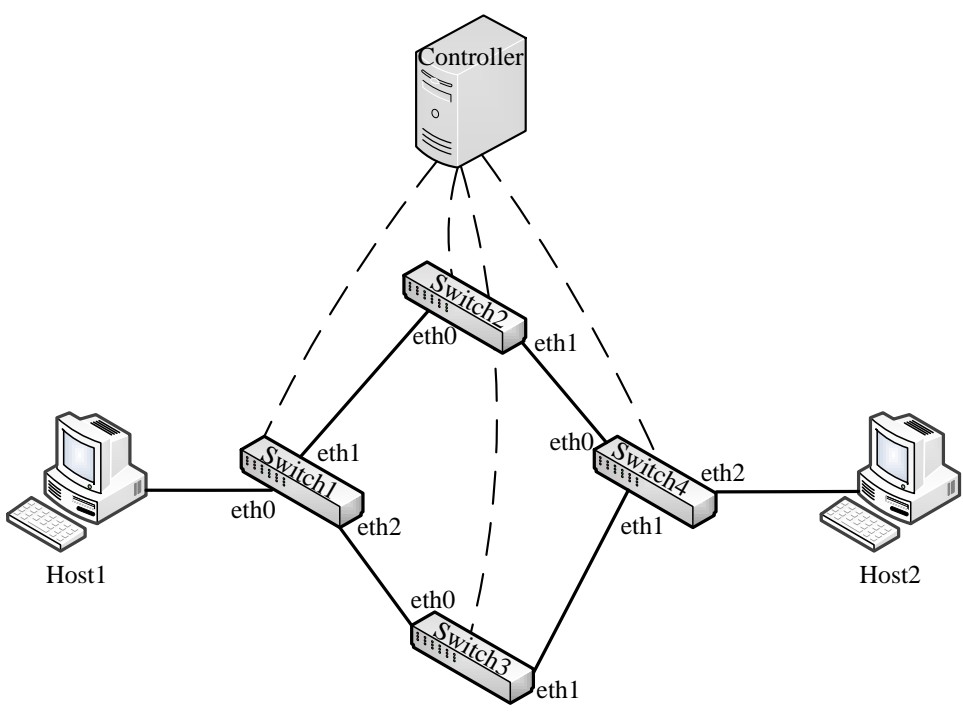

Figure 8  Experimental topology.

the attack model summarized and objectives proposed in the 'Problem description' section. We constructed five types of dataflow, Flow A~Flow E, at Host1 in the network topology shown in Fig. 8. Flow A represents the data flow sent by the user whose attribute feature set is R, Flow B represents the data flow sent by the user whose attribute feature set is P, Flow C represents the data flow sent by the user whose attribute feature set is Q, Flow D represents the data flow obtained after tampering with the data content of Flow B, Flow E represents the data flow obtained by forging the matches of Flow C, assume that Flow E replaces Att_Features of Flow C with Att_Features of Flow B. AISCF establishes access control over data flows flowing into the network, and stipulates that the data flow whose attribute feature set is P and Q can be transmitted in the network. And packets with attribute characteristics P and Q are specified to be transmitted in different paths, assume that when packets are sent from Host1 to Host2, the forwarding path with attribute P and Q are Host1 → Switch1 → Switch2 → Switch4 → Host2 and Host1 → Switch1 → Switch3 → Switch4 → Host2 respectively. To verify the results, the sampling parameter was set as $\theta = 0$ in the effectiveness verification stage; that is, all the packets were sampled, and signature verification based on attribute identification was performed for all of them. Host1 successively sent the five types of packets to Host2 at the rate of 50 packets per second; this step was repeated 10 times. The average was obtained,

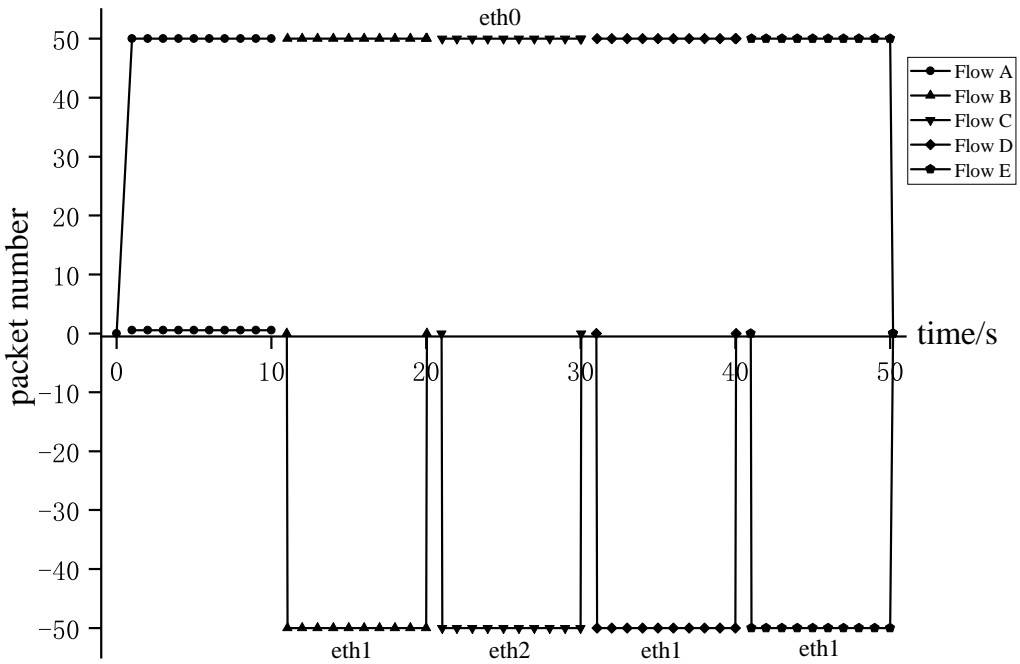

**Figure 9  Switch1 port traffic.**

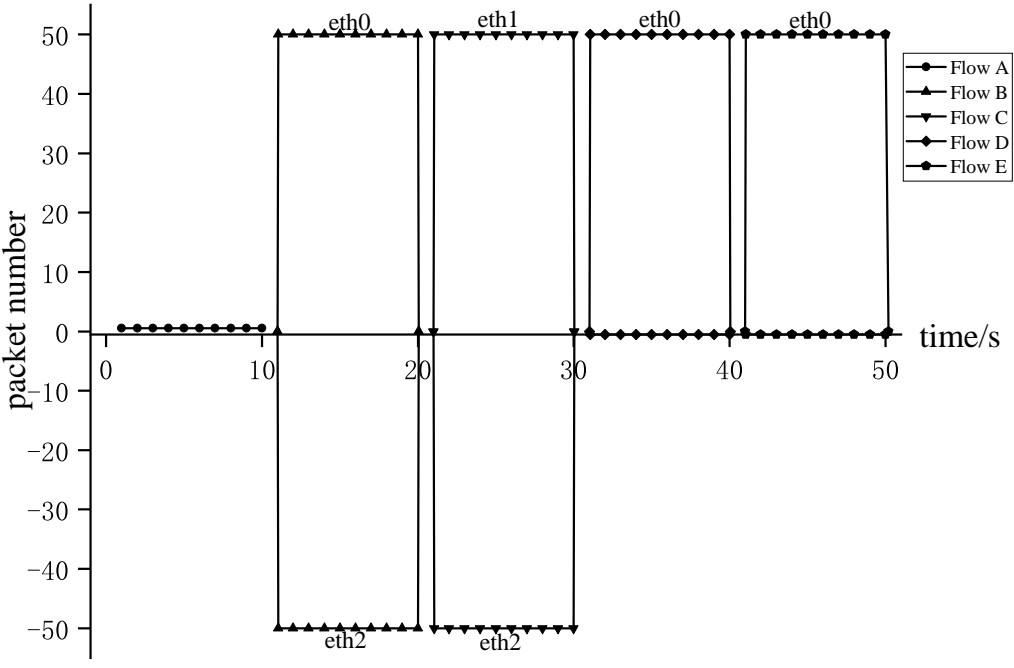

**Figure 10  Switch4 port traffic.**

and the data traffic was measured at the switch port. Figs. 9 and 10 show the traffic at each port of Switch1 and Switch4, respectively, during the experiment.

Figures 9 and 10 show the results for every 10s, representing the forwarding of one type of dataflow on Switch1 and Switch4. The $x$-axis and $y$-axis represent the time and number of packets, respectively. The positive and negative directions of the $y$-axis respectively represent the number of packets received and sent by the port. Flow A received from port eth0 of Switch1 but did not send from any port of Switch1, because Flow A does not meet the access control specified by AISCF and cannot generate attribute identification, the ingress switch in AISCF will discard Flow A, indicating that AISCF can achieve effective access control over data flows. Unlike Flow A, Switch4 sent Flow B and Flow C. This indicates that AISCF can implement fine-grained access control on Flows A to C according to access control rules. Port eth1 of Switch1 sent Flow B, and port eth0 of Switch4 received it. In contrast, port eth2 of Switch1 sent Flow C, and port eth1 of Switch4 received it. These actions comply with the forwarding rules that AISCF formulated for data flows, demonstrating that AISCF can control and forward data flows based on attribute identification. Switch4 received Flows D and E but did not forward them. This indicates that AISCF can detect and block abnormal packets with content tampering and matches forgery through signature verification based on attribute identification.

**Experiment 2**: In Experiment 2, to verify the impact of $\theta$ on AISCF packet verification, we set $\theta$ as 0, 1, 3, 4, 7, and 9. These values correspond to a sampling probability of 100%, 50%, 25%, 20%, 12.5%, and 10%, respectively. Test the performance of AISCF using different $\theta$ in detecting abnormal packets. It is considered as a successful detection when AISCF blocks the data flow which has the same Att_Features item as the abnormal packet's within the specified time, and the false negative of detection is defined as the percentage of undetected times in the total detection times. Host1 continuously sends the security data flow Flow B in Experiment 1 to Host2, and mixes 10% abnormal data flows into Flow B. The abnormal data flows include Flow A, Flow D and Flow E, which correspond to three types of abnormal data flows: the illegally accessed data flow, the data flow with data tampering, and the data flow with forged matching items. The false negative is used as the evaluation parameter for AISCF detection performance to analyze the impact of $\theta$ on AISCF. The results are shown in Fig. 11.

Figure 11 compares the false negatives of abnormal dataflow detected by the AISCF architecture for different sampling factors $\theta$. It can be seen from the figure that AISCF can achieve accurate detection of the illegally accessed data flow, and the false negative rate is 0% which does not change with $\theta$. When $\theta$ was fixed, the false negatives of data tampering and matches forging cases were similar because both malicious flows would change the attribute signature of the packet and could not pass the signature verification based on attribute identification. When the sampling factor is 5, AISCF detects data tampering and matches forging anomalies with false negative rates of 6.5% and 6.6% respectively; when the sampling factor is 3, AISCF obtains false negative rates of 4.5 and 4% for data tampering and matches forging respectively; and when the sampling factor is 0, AISCF can get a detection false negative rate of 0%. It can be concluded that the false negative rate is proportional to the sampling factor, that is, the smaller the sampling factor, the lower

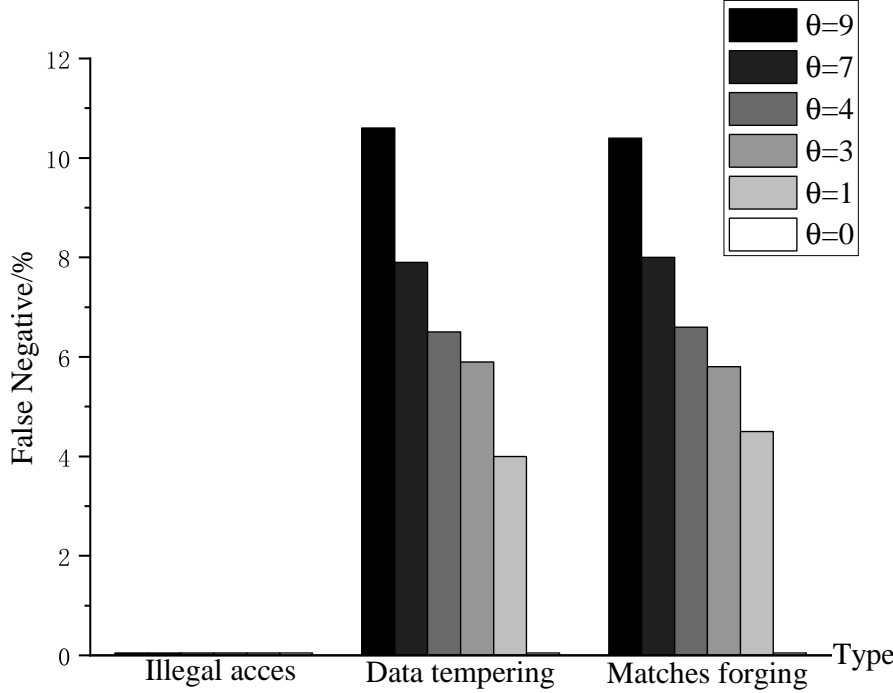

**Figure 11  False negatives in detection.**

the false negative rate of malicious flow detection. Depending on the current network conditions, it is possible to lower $\theta$ to enhance the detection effect of AISCF.

Based on the findings from Experiment 1 and Experiment 2, it is evident that AISCF is capable of effectively detecting malicious activities in the network, such as unauthorized access, data tampering, and spoofing of match items, which ensures the secure transmission of data from end-to-end within an IoT fog architecture. Furthermore, AISCF provides flexible and fine-grained control over data flow, significantly enhancing the efficiency of data management within the IoT.

## Performance evaluation

This section evaluates the network performance of AISCF in terms of packet forwarding delay, network throughput and controller CPU usage.

**Experiment 3**: We conducted Experiment 3 to analyze the time consumption of the ABS scheme in AISCF. We established multiple attribute policy tree structures with the number of attribute features as 5, 10, 15, 20, 25, and 30. The AISCF's time consumption of the signature phase and signature verification phase under various quantities of attribute features was calculated repeatedly for averaging. The results are shown in Fig. 12.

Figure 12 shows that the time consumed by the signature phase is proportional to the number of attributes in the attribute policy tree. The time consumption of the signature phase increases by an average of 0.5 ms for every five attribute features added. A normal local area network (LAN) requires a signature policy composed of 20–30 attribute features.

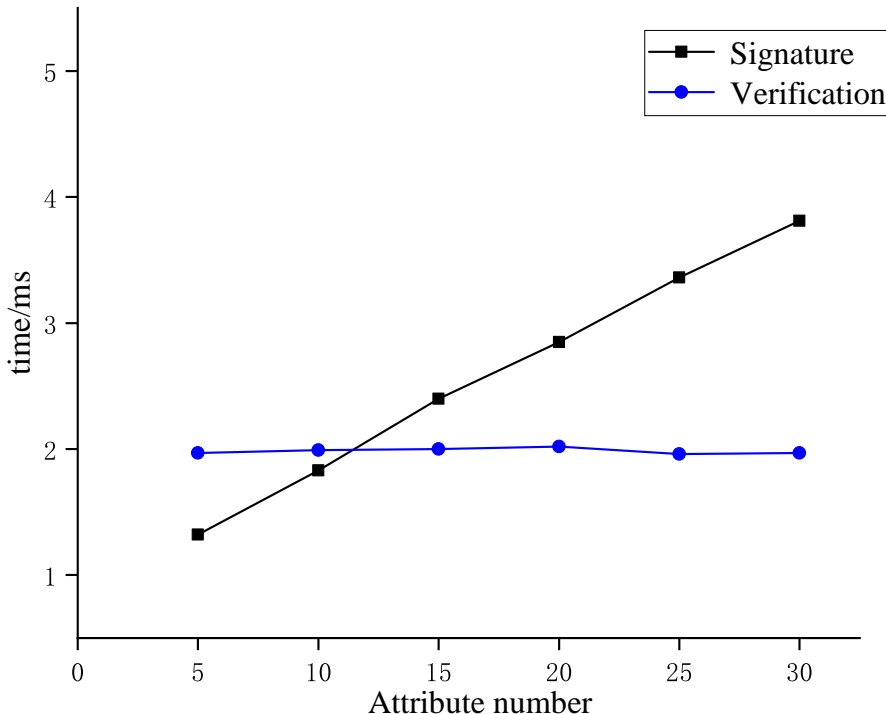

**Figure 12**  **Time consumption of the signature scheme.**

Therefore, when creating packets based on attribute identification, the increase in the time consumption caused by the signature phase at the IoT device side can be controlled within 4 ms, which is an acceptable transmission delay in the AISCF architecture. The average time consumption of the signature verification process is 1.96 ms and is independent of the scale of the signature policy. Therefore, the AISCF architecture can ensure a low and stable time delay of dataflow in the transmission process.

**Experiment 4**: We conducted Experiment 4 to test the dataflow forwarding delay in the AISCF architecture. A total of 1,000 conventional packets and 1,000 packets containing legal attribute identification were created at Host1 and sent to Host2. The conventional packets were transmitted in the basic network without the AISCF architecture, whereas the packets containing attribute identification were transmitted in the network with the AISCF architecture. The sampling factor in the AISCF architecture was set as 7. We captured the packets at Switch1-eth0 and Switch1-eth2, respectively, and calculated the forwarding delay. The experimental results were used to generate the cumulative distribution function (CDF) curves, as shown in Figures 13~15.

Figures 13 and 14 show the forwarding delays of 1,000 packets in the conventional network and AISCF, respectively. The average forwarding delay for 1,000 packets in the basic network was 2.56 ms, whereas the average forwarding delay in the AISCF architecture was 2.95 ms. Using AISCF architecture will increase the forwarding delay by 15.2% during packet forwarding compared to conventional network and the increased forwarding delay originated from two aspects. First, the packets transmitted in the AISCF architecture

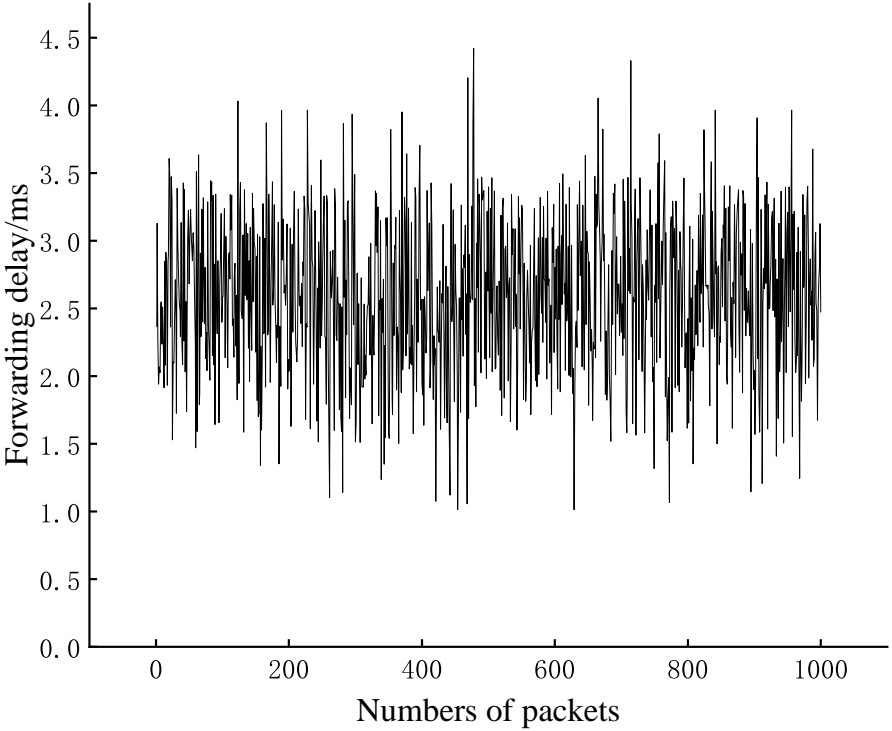

**Figure 13** **Packet forwarding delay in conventional network.**

include an attribute identification field compared with the conventional packets. Second, the AISCF architecture samples the packets and copies the sampled packets to the controller for signature verification process.

Figure 15 plots the cumulative distribution function (CDF) curve of the experimental results, and from the figure we can see that, when packets were transmitted in the basic network, 96% of the packets could be forwarded within 3 ms. When we use the AISCF architecture, 87% of the packets can be forwarded within 3 ms, and 99% of the packets can be forwarded within 5.5 ms. AISCF causes a small increase in the forwarding latency of packets.

Table 4 compares the forwarding delay of data packets between AISCF and related schemes. LPV (*Wang, Li & Zhang, 2019*) has a low verification overhead, but it requires sampling at the first and last switches where the data flow enters and exits, resulting in an increase in forwarding delay. AISCF architecture only samples and verifies data packets at the egress switch, effectively reducing the burden on the controller and the forwarding delay of data packets. P4Label (*Zuo et al., 2020*) performs signature verification in P4 switching devices, thus generating a high verification overhead. AISCF only performs sampling at the switching devices, and delegates the signature verification process to the controller. Attribute-Guard (*Zhu et al., 2020*) has a large time overhead in the verification process, because it involves multiple bilinear pairings and polynomial calculations in the signature verification process, and the number of calculations is linearly related to the

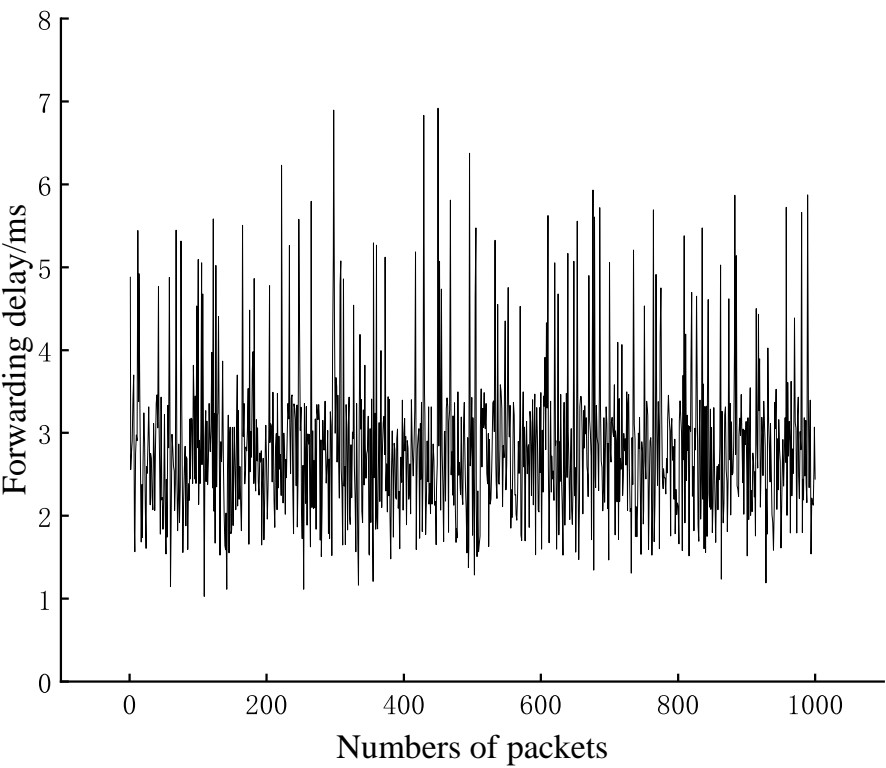

**Figure 14** **Packet forwarding delay in AISCF.**

number of attributes, which increases the forwarding delay of data packets. Compared with Attribute-Guard, AISCF only performs one bilinear pairing and one exponentiation in the signature verification process, which significantly improves the signature verification overhead. Scheme (*Xie et al., 2019*) applies blockchain to 5G Vehicle *Ad-hoc* Network (5G-VANET) and verifies data through a consensus mechanism between multiple nodes. However, data transmission between two points requires confirmation from other nodes. Therefore, when the data volume is large, it can cause an increase in node burden and result in high forwarding latency. Compared to Scheme (*Xie et al., 2019*), AISCF's forwarding latency is only related to the forwarding path and is not affected by other nodes in the network.

**Experiment 5**: The egress switch mirrors and transmits the sampled packets to the controller for signature verification, which affects the throughput of the network. To test the relationship between sampling factor $\theta$ and throughput in AISCF, Experiment 5 is constructed. In Experiment 5, $\theta$ was varied from 0, 1, 3, 4, 7, to 9, and the packets sent in the network were security packets with a load length of 1,000B. Test the throughput of AISCF for many times, take the average value and compare it with the network throughput without AISCF. The result is shown in Fig. 16.

In Fig. 16, the bars show the throughput of the network for different $\theta$, and the line shows the ratio of the throughput of the network with AISCF architecture to the throughput of

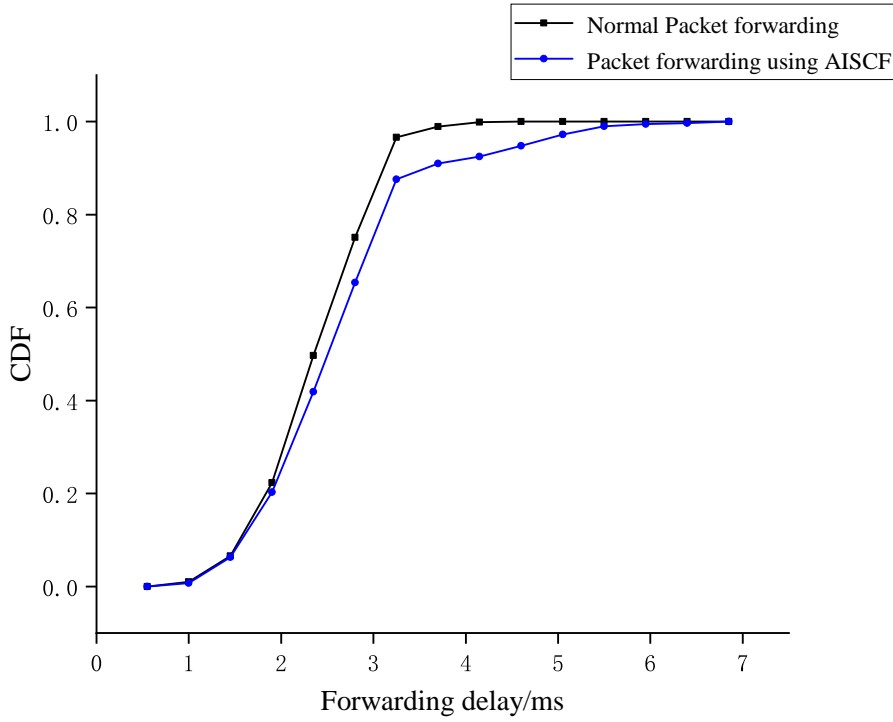

**Figure 15** Forwarding delay CDF curves.

**Table 4** Comparison of packet forwarding latency among different schemes.

| Scheme | Verification scheme | Verification overhead | Forwarding delay | Functions |
|---|---|---|---|---|
| LPV (*Wang, Li & Zhang, 2019*) | Hash-based message authentication code (HMAC) encryption | 0.15 ms | 33.17 ms (3–5 switches) | Detecting and locating forged and tampered packets |
| P4Label (*Zuo et al., 2020*) | Identity-based signature | 16.2 ms | 1.12 ms (3 switches) | Detecting forged and tampered packets |
| Attribute-Guard (*Zhu et al., 2020*) | Attribute-based group signature | 20.2 ms | 30.95 ms (3 switches) | Access control and validity authentication of data flows |
| Scheme (*Xie et al., 2019*) | Blockchain-based verification | Not analyzed | 0.5 s-0.9 s (distance 200 m–500 m) | Node identity authentication and data access control |
| AISCF | Attribute-based signature | 1.96 ms | 2.95 ms (3 switches) | Detecting and blocking forged and tampered packets; fine-grained packet control; tracing abnormal packets |

the basic network. With the decrease in $\theta$, the throughput decreased, when $\theta$ was 0, the throughput was the lowest, which decreased by 42.5% compared with the throughput of the basic network, and when $\theta$ was greater than 7, the network throughput decreased by less than 10% compared with the basic network, which had little impact on the network

performance. Combined with the results of Experiment 2, the sampling factor will affect the detection effect and system performance, such as the network throughput, of AISCF at the same time. AISCF can dynamically select $\theta$ to meet the requirements between the detection effect of abnormal data flow and the system performance.

**Experiment 6**: To evaluate the verification overhead of the dynamic sampling method in AISCF, Experiment 6 is conducted, where Host1 continuously sent secure data flows to Host2. We measured the CPU utilization of the controller in AISCF using both dynamic sampling and fixed sampling factor methods, and compared them with a conventional SDN network. The results are shown in Fig. 17. We defined the function $f(\theta) = 2\theta$ to dynamically select the sampling factor, the maximum sampling number threshold $K = 1000$, and the fixed sampling factor $\theta_0 = 1$.

Figure 17 shows the statistics of the controller's CPU utilization at different time points during data flow transmission. The figure suggests that the average CPU utilization of the controller is 9.21% when the data flow is transmitted in the SDN network. When AISCF uses a fixed sampling rate, the CPU utilization of the controller is stable, with an average value of 18.24%, which is significantly higher than that of the SDN network. When AISCF uses a dynamic sampling rate, the CPU utilization of the controller exhibits a gradual decline, with average values of 16.7%, 14.1%, and 10.3% when the number of packets sent ranges from 2,000 to 5,000, 5,000 to 10,000, and 10,000 to 18,000, respectively. When the number of packets sent ranges from 10,000 to 18,000, AISCF using a dynamic sampling rate reduces the controller's CPU utilization by 44.3% compared to using a fixed sampling rate, indicating that AISCF using a dynamic sampling rate can effectively reduce the verification overhead of the controller for packets by adjusting the sampling factor. AISCF using a dynamic sampling rate increases the controller's CPU utilization by 11.35% compared to SDN network, and the additional CPU resources are used for verifying signatures of sampled packets, tracking malicious users, and other operations. The impact of AISCF on the controller's performance is minimal.

The results of Experiment 3 to Experiment 6 demonstrate that AISCF achieves its security functions with low network performance overhead, which is a feasible and lightweight data flow forwarding verification scheme.

## CONCLUSION

To address the security issues of data lacking effective access control mechanisms and secure verification in the IoT fog architecture, we propose the AISCF, a method of data security control and forwarding based on attribute identification. AISCF applies attribute signatures and SDN to the data packet forwarding verification process in the IoT, achieving fine-grained access control of data, detection and defense of attacks such as data tampering and match item spoofing, and tracing of abnormal data, ensuring the secure transmission of data from end-to-end within the IoT fog architecture. AISCF uses attribute identifiers as match items to match and forward data streams, effectively improving the granularity of data stream management in the IoT. At the same time, AISCF uses a dynamic sampling method to sample and verify data streams, effectively reducing the verification overhead of AISCF. Finally, AISCF is experimentally verified in a simulated IoT fog architecture

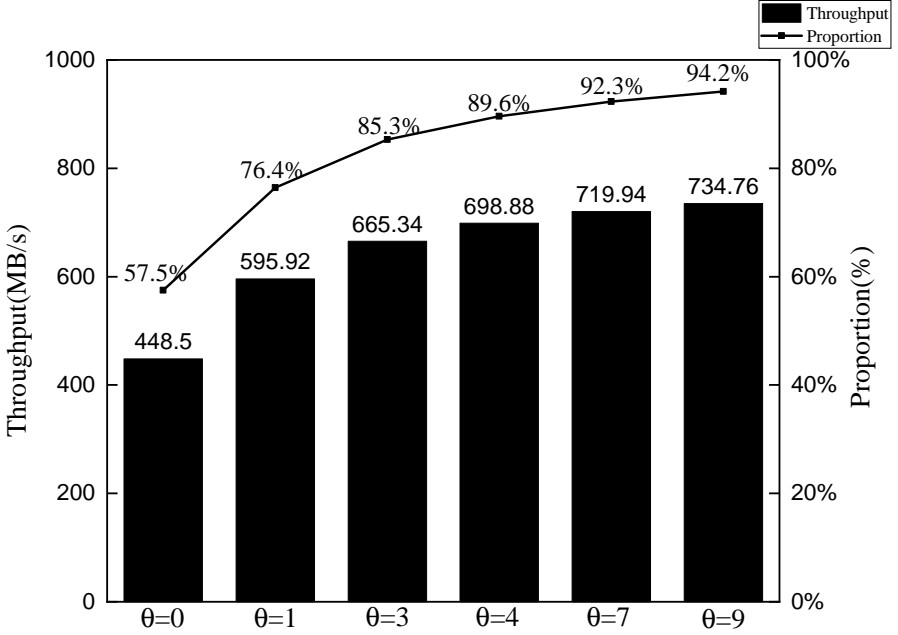

**Figure 16  Throughput analysis.**

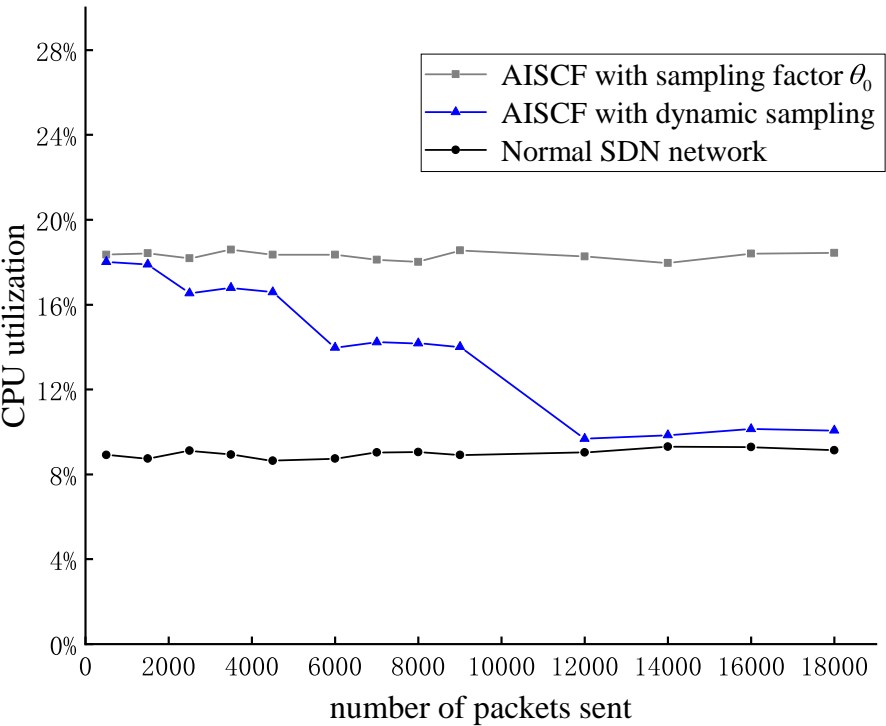

**Figure 17  CPU utilization of the controller.**

environment. The experimental results show that AISCF can effectively ensure the secure transmission of data from end-to-end, and that AISCF has low overhead in terms of data packet forwarding delay, throughput and CPU utilization. AISCF is a lightweight data forwarding verification scheme that has practicality in the IoT fog architecture.

The next step in our research will focus on addressing the issue of detecting and locating malicious nodes in the IoT fog architecture. We aim to establish a transmission path generation scheme based on security scores between IoT devices and fog nodes. This will enable the rapid replacement of secure alternative paths when abnormal data forwarding is detected.

### Funding

This work was supported by the National Natural Science Foundation of China (No. 61572517). The funders had no role in study design, data collection and analysis, decision to publish, or preparation of the manuscript.

### Grant Disclosures

The following grant information was disclosed by the authors:
National Natural Science Foundation of China: 61572517.

### Competing Interests

The authors declare there are no competing interests.

### Author Contributions

- Jingxu Xiao conceived and designed the experiments, performed the experiments, analyzed the data, performed the computation work, prepared figures and/or tables, and approved the final draft.
- Chaowen Chang analyzed the data, authored or reviewed drafts of the article, and approved the final draft.
- Ping Wu performed the experiments, authored or reviewed drafts of the article, and approved the final draft.
- Yingying Ma performed the computation work, authored or reviewed drafts of the article, and approved the final draft.

### Data Availability

The method for generating the verification data in the experimental section of the article is available in the Supplementary File.

### Supplemental Information

Supplemental information for this article can be found online at http://dx.doi.org/10.7717/peerj-cs.1747#supplemental-information.

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
