# Peer review of "Attribute identification based IoT fog data security control and forwarding"

_PeerJ Computer Science, doi:10.7717/peerj-cs.1747_

## Round 0.1 · original submission · Minor Revisions

The paper introduces AISCF, a security framework for IoT fog data control and forwarding, with commendable clarity and mathematical rigor. However, there's a need for clearer differentiation from prior work, additional evaluation metrics, and grammar issues to be addressed. Thematic overlaps with previous work should be clarified. Overall, reviewers appreciate the paper's contributions and rigorous experimental design, which yields impressive results.

**Language Note:** The Academic Editor has identified that the English language must be improved. PeerJ can provide language editing services - please contact us at copyediting@peerj.com for pricing (be sure to provide your manuscript number and title). Alternatively, you should make your own arrangements to improve the language quality and provide details in your response letter. – PeerJ Staff

Reviewer 1 ·

Basic reporting

English is used clearly in this article.

Literature references, sufficient field background/context is provided. In the introduction part, the authors introduce the background knowledge of IoT-Fog and the security threats it faces. Then the authors propose an attribute identification based security control and forwarding method for IoT fog data.In the related work part, the authors list a large volume of existing works to discuss the current security defense methods and their disadvantages.

Professional article structure, figures, tables. Raw data packets are shared.

Experimental design

In the evaluation part, the authors implement their tool using P4 programmable software switches, Several experiments are conducted to answer different research questions. For validity analysis, the authors validate that their implementation can detect abnormal packets. In performance evaluation, the authors evaluates the network performance of their tools in terms of packet forwarding delay, network throughput and controller CPU usage. The phenomenon are depicted as figures and explained.

Validity of the findings

The experimental results show that the tool can effectively ensure the secure transmission of data from end-to-end, and that the tool has low overhead in terms of data packet forwarding delay, throughput and CPU utilization.

Cite this review as

·

Basic reporting

The paper introduces the AISCF model, a security framework designed for IoT fog data control and forwarding.The authors provided a clear and concise definition of the Internet of Things (IoT), highlighting its significance in connecting various physical devices and enabling data collection, transmission, and processing. AISCF employs attribute identification to manage data streams, enhancing security in IoT fog architectures. The research evaluates AISCF's effectiveness using P4 programmable software switches in a simulated environment. Experiments demonstrate AISCF's capability to control access, verify security, and fine-tune data flow forwarding. A standout feature is the model's dynamic sampling method, which helps to reduce the verification overhead, paving the way for optimized performance. The study shows AISCF is a accentuates a nimble yet potent solution, poised to redefine data security paradigms in IoT fog systems.
However, the paper's layout warrants some reconfiguration. Currently, all figures are positioned at the end of the paper, akin to an appendix. However, it would enhance the reader's experience if the figures were integrated directly within the main body, precisely at the points of reference.

Experimental design

The article delves into technical aspects, such as the Attribute-Based Signature (ABS) scheme and the AISCF architecture. This depth showcases a thorough understanding of the subject matter.Delving into the mathematical intricacies, the paper is commendable for its rigorous approach. It employs mathematical formulations, such as polynomial equations, Lagrangian coefficients, and bilinear mappings, to describe the AISCF model's workings. These formulations provide a rigorous foundation for the proposed security framework. The paper provides clear definitions for terms and notations used, such as AttID, msk, mpk, and PP. This aids in understanding the mathematical constructs and their relevance to the AISCF model. Furthermore, the paper uses tools like hash functions (H), elliptic curves, and cyclic groups (G1, G2, GT). These tools are standard in cryptographic literature and are essential for ensuring data security.
However, there's a discernible thematic overlap with a prior work titled “A Secure Data Flow Forwarding Method Based on Service Ordering Management.” It would be a better way to draw parallels and distinctions with the previous works.

Validity of the findings

The AISCF model, as presented in the paper, has undergone rigorous testing to validate its effectiveness. The experimental environment was set up using P4 programmable software switches, and Mininet was employed to simulate the IoT fog architecture environment. The experiments were designed to verify AISCF's capabilities in access control, security verification, and fine-grained control forwarding of data flow. The results from the experiments demonstrated that AISCF is adept at detecting malicious activities in the network, such as unauthorized access, data tampering, and spoofing of match items. The dynamic sampling method employed by AISCF reduces the verification overhead, making it more efficient.
The paper's evaluation of AISCF, while comprehensive, could benefit from providing more evaluation metrics. When discussing the performance of AISCF, consider providing more quantitative data, such as benchmarks, latency measurements, or throughput rates. This can offer a more concrete understanding of its efficiency. It would be insightful to compare the performance of AISCF with other existing systems or models. This can highlight the advantages and unique features of AISCF. For instance, there's burgeoning interest in leveraging blockchain to improve privacy for fog computing and internet of thing data. A comparative study elucidating the merits of AISCF with blockchain-based solutions would be a compelling addition to spotlighting AISCF’s uniqueness and advantages.

Reviewer 3 ·

Basic reporting

This paper presents a novel security control and forwarding method for IoT fog data that integrates attribute signatures with the IoT fog layer. By leveraging Software Defined Network, it establishes an access control mechanism rooted in attribute features, enhancing the security and traceability of incoming data flows in the IoT fog layer. The method utilizes attribute identification for SDN control forwarding, facilitating fine-grained control based on attribute signature. Also, the system can identify malicious packets, such as those exhibiting data tampering and forged matching items, ensuring the integrity of the end-to-end data packet forwarding process. While this paper clearly explain the proposed method, it suffers from numerous grammatical issues. These not only hinder a smooth read but also occasionally obscure the intended meaning. It is strongly recommended that the manuscript undergo several rigorous proofreading processes to fix these issues.

Experimental design

The paper clearly defines the problem, which is both important and practical in IOT systems. The experiments reflects a comprehensive and rigorous investigation.

Validity of the findings

The results, as presented, indicate that the proposed approach is efficient in detecting threats like data tampering and forged matching items. The findings on the overhead on network throughput, CPU utilization, and packet forwarding latency are particularly impressive and highlight the practicality of the approach in an IoT system.

Additional comments

There are some grammar issues in the paper. Also, while reviewing your paper, I noticed an overuse of passive voice throughout the paper that makes the writing wordy and a little difficult to follow. I strongly recommend the authors to do multiple rounds of proofreading and rephrase some sentences to utilize the active voice that is more direct and engaging.

Examples:

Wordy: In this section, an experimental environment is built and the validity and performance of AISCF are assessed by conducting experiments.

Grammar issue: Implement the AISCF architecture using P4 programmable software switches, which uses P4 language and generates JSON format description file through p4 compiler(P4c), and imports it into P4 behavioral-model version 2(BMv2) to run.

Grammar issue: To select a suitable sampling method to sample and verify data flows, avoiding generating large verification overhead that would have a significant impact on the data transmission efficiency and network performance.

Cite this review as

---

## Round 0.2 · accepted · Accept

Congratulations! The paper presents AISCF, an IoT fog data control and forwarding security framework, with clarity and mathematical rigor.

Reviewer 3 ·

Basic reporting

The authors have addressed my concerns in the revised manuscript.

Experimental design

no comment

Validity of the findings

no comment

Cite this review as